

1 2 3

# The Effects of Ocean Surface Waves on Global Intraseasonal Prediction: Case Studies with a Coupled CFSv2.0-WW3

Ruizi Shi[1], Fanghua Xu[1*], Li Liu[1], Zheng Fan[1], Hao Yu[1], Hong Li[1,3], Xiang Li[2] and
Yunfei Zhang[2]
[1] Ministry of Education Key Laboratory for Earth System Modeling, and Department of Earth System
Science, Tsinghua University, Beijing, 100084, China
[2] Key Laboratory of Marine Hazards Forecasting, National Marine Environmental Forecasting Center,
Ministry of Natural Resources, Beijing, 100081, China
[3] Department of Atmospheric and Oceanic Sciences and Institute of Atmospheric Sciences, Fudan
University, Shanghai, 200433, China.
*Correspondence to*: Fanghua Xu (fxu@mail.tsinghua.edu.cn)





**Abstract.** Ocean surface gravity waves have enormous effects on physical processes at the atmosphere–
ocean interface. The effects of wave-related processes on global intraseasonal prediction were evaluated
after we incorporated the WAVEWATCH III model into the Climate Forecast System model version 2.0
(CFSv2.0), with the Chinese Community Coupler version 2.0. Several major wave-related processes,
including the Langmuir mixing, Stokes-Coriolis force with entrainment, air-sea fluxes modified by
Stokes drift and momentum roughness length, were evaluated in two groups of 56-day experiments, one
for boreal winter and the other for boreal summer. Comparisons were performed against in-situ buoys,
satellite measurements and reanalysis data, to evaluate the influence of waves on intraseasonal prediction
of sea surface temperature (SST), 2-m air temperature (T02), mixed layer depth (MLD), 10-m wind speed
(WSP10) and significant wave height (SWH) in CFSv2.0. Overestimated SST and T02, as well as
underestimated MLD in mid and high latitudes in summer from original CFSv2.0 are clearly improved,
mainly due to enhanced vertical mixing generated by Stokes drift. The largest regional mean SST
improvement reaches 35.89% in the Southern Ocean. For WSP10 and SWH, the wave-related processes
generally lead to reduction of biases in regions where wind speed and SWH are overestimated. The
decreased SST caused by Stokes drift-related mixing stabilizes marine atmospheric boundary layer,
weakens wind speed and then SWH. Compared with the NDBC buoy data, the overestimated WSP10 is
improved by up to 13.52% in boreal summer. The increased roughness length due to waves leads to some
reduction in the originally overestimated wind speed and SWH, with the largest SWH improvement of
11.93% and 20.05% in boreal winter and summer respectively. The effects of Stokes drift and current on
air-sea fluxes are investigated separately. Their overall effects on air-sea fluxes reduce the overestimated
WSP10 by up to 17.31% and 23.21% in boreal winter and summer respectively. These cases are helpful
for the future development of the two-way CFS-wave coupled system.



## 1 Introduction

Ocean surface gravity waves play an important role in modifying physical processes at the
atmosphere–ocean interface, which can influence momentum, heat and freshwater fluxes across the air-
sea interface (Li and Garrett 1997; Taylor and Yelland, 2001; Moon et al., 2004; Janssen 2004; Belcher
et al., 2012; Moum and Smyth, 2019). For instance, ocean surface waves modify ocean surface roughness
to influence the marine atmospheric boundary layer and thus change the momentum, latent heat, and
sensible heat transfer (Janssen 1989, 1991; Taylor and Yelland, 2001; Moon et al., 2004; Drennan et al.,
2003, 2005). The breaking waves inject turbulent kinetic energy in the upper ocean, which enhances the
mixing process (Terray et al. 1996). Nonbreaking surface waves also affect mixing in the upper ocean
by adding a wave-related Reynolds stress (Qiao et al., 2004; Ghantous and Babanin, 2014). The wave-
related Stokes drift interacts with Coriolis force and produces the Coriolis-Stokes force (Hasselmann
1970). The shear of Stokes drift is critical for generation of Langmuir circulation, which significantly
deepens the mixed layer by strong vertical mixing both at climate scales (Li and Garrett 1997; Belcher
et al., 2012) and at weather scales (Kukulka et al., 2009).
As Fox-Kemper et al. (2019) indicated, the improvement to atmosphere-ocean coupling with a better
representation of the effects of surface gravity waves, is one of the challenges and focuses in ocean
modeling for the next decade. Regional coupled models were developed to study tropical cyclones, storm
surge and other coastal processes at small or meso scales (e.g., Prakash et al., 2018; Ricchi et al., 2017;
Pianezze et al., 2018; Wu et al., 2019). The Coupled Ocean-Atmosphere-Wave-Sediment Transport
Modeling System (COAWST) developed by Warner et al. (2010) is one of the well-known fully-coupled
regional models, which has been applied in various locations such as the South China Sea (Sun et al.,
2019; Wu et al., 2019), the Bay of Bengal (Prakash et al., 2018) and the Mediterranean Sea (Ricchi et al.,



2017). On the other hand, most of the coupled models with a wave component at global scale were
developed for climate research (e.g., Qiao et al., 2010; Law-Chune and Aouf, 2018; Fan et al., 2012; Fan
and Griffies, 2014; Li et al. 2016, 2017). Exceptionally, an Integrated Forecasting System (IFS) with
fully coupled atmosphere, ocean and wave components, developed by European Centre for Medium-
Range Weather Forecasts (ECMWF) (Janssen 2004; Bidlot et al. 2019, 2020), has been released with
great flexibility for global forecasts from medium-range weather scales to seasonal scales (Breivik et al.

64  2015).

The overall effects of wave-related processes on numerical prediction have been shown to be important
in coupled systems (e.g., Law-Chune and Aouf, 2018; Bao et al. 2019; Couvelard et al. 2020). The
contribution of individual wave-related process, however, is complex and studied individually (e.g.
Janssen 2004; Breivik et al. 2015; Janssen and Bidlot 2018; Pineau-Guillou et al. 2018), and worth further
evaluation in different modelling systems. Since it takes sufficient periods for the wave energy to develop
(Janssen 2004), it is of great interest to investigate the impact of individual wave effect at intraseasonal
timescale in a new global atmosphere-ocean-wave system. To achieve this, we coupled the
WAVEWATCH III (WW3) to the Climate Forecast System model version 2.0 (CFSv2.0) and then
conducted sensitivity experiments in boreal winter and summer for comparison. The CFSv2.0 is a
coupled system with main application for intraseasonal and seasonal prediction (e.g. Saha et al. 2014).
The National Centers for Environmental Prediction (NCEP) is establishing its own atmosphere-ocean-
wave system, in which the Global Forecast System (GFS; the atmosphere module in CFS system) is one-
way coupled with WW3. Our work can provide insights for two-way wave coupling of CFS, and is
helpful for the future development of the CFS-wave coupling system.
Several wave-related processes are studied, including upper ocean mixing modified by Langmuir cell,





Stokes-Coriolis force and entrainment, air-sea fluxes modified by surface current and Stokes drift, and
momentum roughness length. All these processes greatly affect momentum and enthalpy fluxes across
the air-sea interface (e.g., Fan et al., 2012; Fan and Griffies, 2014; Li et al. 2016, 2017; Renault et al.
2012; Varlas et al. 2020). Two groups of 56-day predictions were conducted for boreal winter and boreal
summer, respectively. The predictions were then compared with observations and reanalysis data. For
each group, sensitivity experiments with different wave parameterizations were carried out to evaluate
the effects of individual wave-related process. The rest of the paper is structured as follows: methods and
numerical experiments with different parameterizations are described in Section 2; the observations and
reanalysis data are introduced in Section 3, and the results of experiments are evaluated and compared
in Section 4. Finally, a summary and discussion are given in Section 5.
**2 Methods and Experiments**
**2.1 Coupling WAVEWATCH III with CFSv2.0**
The version 5.16 of WW3 (WAVEWATCH III Development Group, 2016) developed by the National
Oceanic and Atmospheric Administration (NOAA)/NCEP has been incorporated into the CFSv2.0 (Saha
et al., 2014) as a new model component. The latitude range of WW3 is 78°S–78°N with a spatial
resolution of 1/3°; the frequency range is 0.04118-0.4056Hz and the total number of frequencies is 25;
the number of wave directions is 24 with a resolution of 15°; the maximum global time step and the
minimum source term time step are both 180 s.
The CFSv2.0 contains two components, the GFS (details are available at
http://www.emc.ncep.noaa.gov/GFS/doc.php) as the atmosphere component and the Modular Ocean
Model version 4 (MOM4; Griffies et al., 2004) as the ocean component. The MOM4 is integrated on a



nominal 0.5° horizontal grid with enhanced horizontal resolution to 0.25° in the tropics, and has 40
vertical levels; the vertical spacing is 10 m in the upper 225 m, and then increases in unequal intervals to
the bottom at 4478.5 m. The GFS uses a spectral triangular truncation of 382 waves (T382) in the
horizontal, which is equivalent to a grid resolution of nearly 35 km, and 64 sigma-pressure hybrid layers
in the vertical. The time steps of both MOM4 and GFS are 180 s. The ocean and atmosphere components
are then coupled at the same rate. In the two-way coupled system, the GFS receives SST from MOM4
and sends fluxes of heat, momentum, freshwater to MOM4.
The Chinese Community Coupler version 2.0 (C-Coupler2; Liu et al., 2018) is applied to interpolate
and pass variables between atmosphere and wave components as well as ocean and wave components.
Each component receives inputs and supplies outputs on its own grids. The C-Coupler2 is a common,
flexible and user-friendly coupler, which contains a dynamic 3-D coupling system and enables variables
to remain conserved after interpolation. From a series of tests of coupling experiments, the time step of
wave coupling (1800 s) was selected to compromise time consumption and model bias (details in Table
S1 of the supplementary).
A schematic diagram of the coupled atmosphere-ocean-wave system is shown in Fig. 1. As illustrated,
WW3 is two-way coupled with MOM4 and GFS, through the C-Coupler2. WW3 is forced by 10-m wind
from GFS, and then generates and evolves the wave action density spectrum. Meanwhile, the surface
Stokes drift velocity with turbulent Langmuir number is passed to MOM4 (see Section 2.3) from WW3,
and the surface Stokes drift velocity and the Charnock parameter are passed to GFS (see Section 2.4 and
2.5). The high frequency tail assumption for Stokes drift in WW3 is used with a spectral level decaying
as $f^{-5}$ (frequency). Additionally, the ocean surface current velocities from MOM4 are also passed to GFS
(see Section 2.4). In this study, both the CFS and WW3 use warm starts; the daily initial fields at 00:00





UTC for CFS were generated by the real time operational Climate Data Assimilation System (Kalnay et
al., 1996), downloaded from the CFS official website
(http://nomads.ncep.noaa.gov/pub/data/nccf/com/cfs/prod). To get initial conditions for WW3, a stand-
alone WW3 model is set up synchronously (see Section 2.2).
**2.2 Initialization of WAVEWATCH III**

In WW3, input of momentum and energy by wind, and dissipation for wave-ocean interaction are two

important terms (combined as input-dissipation source term) in the energy balance equation
(WAVEWATCH III Development Group, 2016), which include the Charnock parameter related
estimation. Several different packages to calculate the input-dissipation source term (ST) are offered in
the WW3 version 5.16, including ST2 (Tolman and Chalikov, 1996), ST3 (Janssen, 2004; Bidlot, 2012),
ST4 (Ardhuin et al., 2010), and ST6 (Zieger et al., 2015).

The initial wave fields were generated from 10-day simulation starting from rest in a stand-alone WW3

model. To minimize the biases of initial wave fields, we tested simulations with ST2, ST3, ST4, and ST6
schemes respectively, and compared the results with Janson-3 observations. Two 10-m wind datasets,
the Cross-Calibrated Multi-Platform (CCMP; Atlas et al., 2011) data and the fifth generation European
Centre for Medium-Range Weather Forecasts (ECMWF) Reanalysis (ERA5; Hersbach et al., 2020) data,
were used to drive the wave model respectively. Compared all results, the ST4 scheme with ERA5 wind
forcing generates the minimum significant wave height (SWH) bias (Table S2 in the supplementary),
consistent with findings in Stopa et al. (2016). Thus, the ST4 scheme was chosen to calculate the input
and dissipation term, and generate initial wave fields with ERA5 wind forcing for experiments listed in
Table 1. The parameters used for ST4 scheme followed TEST471f from WAVEWATCH III





Development Group (2016), which is the CFSR (CFS Reanalysis) tuned setup and is commonly-used at
global scale.

**2.3 Parameterizations of Stokes Drift-Related Ocean Mixing**

**2.3.1 Mixing of Langmuir Turbulence**

McWilliams and Sullivan (2000) modified the turbulent velocity scale W in KPP by introducing an
enhancement factor $\varepsilon$, to account for both boundary layer depth changes and nonlocal mixing by
Langmuir turbulence. Based on their work, Van et al. (2012) improved the enhancement factor
corresponding to alignment and misalignment of winds and waves. Li et al. (2016) evaluated these
parameterizations in a coupled global climate model, and parameterizations from Van et al. (2012)
showed best performance. However, the difference between parameterizations with alignment and
misalignment was not significant, owing to the coarse resolution which cannot accurately represent the
refraction by coasts and current features. Besides, the misalignment will certainly increase the runtime
due to increased variables to be transferred from wave to ocean. Hence, we employed the
parameterization corresponding to alignment of winds and waves from Van et al. (2012).
In the study, W (W=$ku_*/\phi$, where $u_*$ is the surface friction velocity, $\phi$ is the dimensionless flux
profile, and $k$=0.4 is the von Kármán constant) varies in proportion to the turbulent Langmuir number,
that is,

$$W = \frac{ku_*}{\phi}\varepsilon, \qquad (1)$$

$$\varepsilon = \sqrt{1 + (3.1La_t)^{-2} + (5.4La_t)^{-4}}, \qquad (2)$$

where $La_t$ is the turbulent Langmuir number, defined as





$$La_\text{t} = \sqrt{\frac{u_*}{|u_s(0)|}}, \tag{3}$$

with $u_s(0)$ is the surface Stokes drift velocity.
Furthermore, the enhanced W will influence the calculation of boundary layer depth. In KPP the
boundary layer depth is determined as the smallest depth at which the bulk Richardson number equals
the critical value $Ri_\text{cr} = 0.3$, that is,

$$Ri_b(h) = \frac{gh[\rho_r - \rho(h)]}{\rho_0[|u_r - u(h)|^2 + W^2]} = Ri_\text{cr}, \tag{4}$$

where $g$ is acceleration of gravity, $\rho$ is density, $u$ is velocity, $\rho_r$ is surface density, $u_r$ is surface
velocity, $\rho_0$ is an average value and $h$ is the boundary layer depth. Hence, when W is enhanced, the
boundary layer depth $h$ is deepened accordingly.
**2.3.2 Stokes–Coriolis Force and Associated Entrainment**
Because the Stokes drift velocity is an increment superimposed on the original current velocity, the
Coriolis force and the Stoke drift together produce an additional so-called Stokes–Coriolis (SC) force
(Hasselmann 1970), that is,

$$SC\ Force = \vec{u_s} \times f\vec{z}. \tag{5}$$

Here $\vec{u_s}$ is surface Stokes drift velocity vector, $f$ is the Coriolis frequency, and $\vec{z}$ is the vertical unity
vector.
To depict the entrainment below the ocean surface boundary layer induced by Stokes drift, Li et al.
(2016) suggested to add the square of surface Stokes drift velocity ($|u_s(0)|^2$) to the denominator of Eqn.
4, that is,

$$Ri_b(h) = \frac{gh[\rho_r - \rho(h)]}{\rho_0[|u_r - u(h)|^2 + W^2 + |u_s(0)|^2]} = Ri_\text{cr}. \tag{6}$$



The boundary layer depth $h$ in KPP from Eqn. 6 is then enhanced due to Stokes drift velocity.
**2.4 Stokes Drift and Sea Surface Current on Air–Sea Fluxes**
At air-sea boundary layer, the momentum flux ($\tau$), sensible heat flux (SH) and freshwater flux (E) are
calculated as

$$\tau = \rho_a C_d |\Delta\vec{V}|\Delta\vec{V}, \qquad (7)$$

$$SH = \rho_a C_h |\Delta\vec{V}|\Delta\theta, \qquad (8)$$

$$E = \rho_a C_e |\Delta\vec{V}|\Delta q, \qquad (9)$$

where $C_d$, $C_h$, $C_e$ are surface exchange coefficients for momentum, sensible heat and freshwater. $\rho_a$
is air density. $\Delta\theta, \Delta q$ are potential temperature and humidity differences between air and sea, and $\Delta\vec{V}$
is velocity of air relative to water flow.
In CFS, $\Delta\vec{V}$ is set to be wind speed ($\overrightarrow{U_{wind}}$). However, the effect of ocean surface current should not
be ignored. Luo et al. (2005) first indicated that including ocean surface current ($\overrightarrow{U_{surf}}$) improves
estimates of $\tau$ and subsequent ocean response. Renault et al. (2016) further indicated that the
improvements of $\tau$ by $\overrightarrow{U_{surf}}$ also feed back into atmosphere. At present, $\Delta\vec{V} = \overrightarrow{U_{wind}} - \overrightarrow{U_{surf}}$ is
widely used in coupled ocean-atmosphere models (e.g., Hersbach and Bidlot, 2008; Takatama et al.,
2017; Renault et al., 2021). Furthermore, Bao et al. (2019) indicated that as a part of the sea surface water
movement with speed magnitude comparable to surface current in mid-high latitudes, the surface Stokes
drift ($\overrightarrow{u_s(0)}$) should also be included, that is,

$$\Delta\vec{V} = \overrightarrow{U_{wind}} - \overrightarrow{U_{surf}} - \overrightarrow{u_s(0)}. \qquad (10)$$

To account for the effects of Stokes drift velocity, the Eqn. 10 was applied in the coupled experiments
(Table 1), and the difference compared with $\Delta\vec{V} = \overrightarrow{U_{wind}} - \overrightarrow{U_{surf}}$ was analyzed in Section 4.4. Note



that the direction of Stokes drift is generally consistent with 10-m wind (Fig. S1a. b, e, f in
supplementary), but the directions of surface current and 10-m wind are usually with an angle due to
Coriolis effect (Fig. S1a-d). Consequently, the effects of $\overrightarrow{U_{surf}}$ and $\overrightarrow{u_s(0)}$ on $\Delta\vec{V}$ depend on the
angles between them and $\overrightarrow{U_{wind}}$.
**2.5 Parameterizations of Momentum Roughness**
In CFS, the fluxes of momentum, heat, and freshwater are passed from atmosphere to ocean, and the
estimates of them are critically important. The fluxes are in part determined by surface roughness length,
which can be converted to surface exchange coefficients in Eqn. 7-9.
**2.5.1 The Momentum Roughness Length in GFS**
In GFS, the momentum roughness length $z_0$ has two terms. The first term $z_{ch}$ is parameterized by
the Charnock relationship (Charnock, 1955) representing wave-resulted sea surface roughness, and the
second term $z_{vis}$ is the viscous contribution (Beljaars, 1994) for low winds and smooth surface, that is,

$$z_0 = z_{ch} + z_{vis} = \frac{C_{ch}u_*^2}{g} + \frac{0.11\nu}{u_*}. \tag{11}$$

Here $C_{ch} = 0.014$ is the constant Charnock parameter, $\nu$ is the air kinematic viscosity. The relation of
$z_0$ in GFS versus 10-m wind speed is shown in Fig.2 (black line).
**2.5.2 The Charnock Relationship Related to Wave State**
When ocean surface waves are explicitly considered, the Charnock parameter $C_{ch}$ is not a constant
(Janssen 1989, 1991; Taylor and Yelland, 2001; Moon et al., 2004; Drennan et al., 2003, 2005). There
are primarily three methods for $C_{ch}$, assessed from the wave-induced kinematic stress (Janssen 1989,





1991), the wave age (Drennan et al., 2003, 2005; Moon et al., 2004; Fan et al., 2012), or the steepness
(Taylor and Yelland, 2001). The former two are based on the wind-sea conditions, whereas the latter
includes both swells and wind-sea waves. In the study, we adopted a method developed by Fan et al.
(2012), which considered the surface roughness leveling off under extremely high wind speed based on
the researches of Powell et al. (2003), Donelan et al. (2004), and Moon et al. (2004). In the
parameterization, $C_{ch}$ is calculated by the wave age $\frac{c_{pi}}{u_*}$ ($c_{pi}$ is the peak phase speed of the dominant
wind-forced waves) as

$$C_{ch} = a(\frac{c_{pi}}{u_*})^b, \tag{12}$$

$$a = \frac{0.023}{1.0568^{U_{10}}}, b = 0.012U_{10}, \tag{13}$$

where $U_{10}$ is the 10-m wind speed. Figure S2 in supplementary shows the $C_{ch}$ distribution obtained by
Eqn. 12-13. Small wind direction variations at low latitudes lead to large wave age and thus high $C_{ch}$,
and vice versa. At mid-high latitudes $C_{ch}$ is higher in summer than in winter.
The relationships between $z_0$ and $U_{10}$ in GFS, WW3 ST4 scheme (Janssen 1989, 1991) and ST4-
FAN scheme (Fan et al., 2012) were compared in Fig.2. The $z_0$ in GFS increases relatively slowly with
increasing wind speed (black). The value of $z_0$ from ST4 scheme (purple) increases rapidly with wind
speed at high winds. In comparison, in ST4-FAN scheme (dark red) the rapid increase of $z_0$ at high
wind speed is obviously restrained, although the mean $z_0$ is slightly higher than that in GFS especially
at wind speed >10 m/s due to larger $C_{ch}$ (>0.014 in Fig. S2). Furthermore, since the Charnock number
is constant in GFS, the standard deviation (STD) of $z_0$ at a given wind speed is near zero. Since the $z_0$
is determined only by wind-sea conditions in ST4 and ST4-FAN scheme, the STD at a given wind speed
is mainly owing to variations in wind fetch (Shimura et al., 2017). The reduced STDs in ST4-FAN
scheme, compared to ST4, imply less sensitivity of $z_0$ to fetch.





**2.6 Set of Experiments**


A series of numerical experiments was conducted to evaluate the effects of aforementioned wave-
related processes on ocean and atmosphere in two 56-day periods, from January 3 to February 28, 2017
and from August 3 to September 28, 2018 for boreal winter and boreal summer, respectively.
The reference experiment (CTRL) is a one-way coupled experiment, in which GFS provides 10-m
wind to WW3, whereas no variable is transferred from WW3 to CFS. The results of CFS in CTRL are
consistent with the corresponding CFS Reanalysis data (Saha et al., 2010). For each period, four
sensitivity experiments were carried out (Table 1). Based on CTRL, the first is VR12-AL-SC-EN
experiment, in which the Langmuir mixing parameterization is applied with Stokes–Coriolis force and
entrainment in MOM4. The second is Z0-FAN experiment, in which the constant $C_{ch}$ in GFS is replaced
by $C_{ch}$ from WW3 ST4-FAN scheme. The effect of fluxes in GFS generated by $\Delta\vec{V}$ (Eqn. 10) is tested
in the FLUX experiment. The last experiment is the ALL, which includes all three parameterizations.

**3 Data**


Due to the availability of in situ and reanalysis data in the simulation periods, only sea surface
temperature (SST), ocean subsurface temperature and salinity (T/S), 2-m air temperature (T02), 10-m
wind speed (WSP10), and significant wave height (SWH) were used to evaluate the simulation results.
The daily average satellite Optimum Interpolation SST (OISST) data were obtained from NOAA, with
0.25°×0.25° resolution (Reynolds et al., 2007; https://www.ncdc.noaa.gov/oisst). The global Argo
observational profiles of T/S (Li et al., 2019) were from China Argo Real-time Data Center
(www.argo.org.cn). The ERA5 datasets of T02, WSP10 and SWH with a spatial resolution of 0.5° were
also used (Hersbach et al., 2020; https://cds.climate.copernicus.eu/cdsapp#!/dataset/reanalysis-era5-





single-levels), which assimilated huge amounts of historical data and thus provided reliable hourly
estimates. Additionally, the WSP10 and SWH observations from the available National Data Buoy
Center (NDBC) buoy data (https://www.ndbc.noaa.gov) were applied for comparison.

**4 Experimental Results**

In this section, an evaluation of simulation results was presented. Comparisons were made between
model results and observations/reanalysis data. The results in the first three days were excluded in the
evaluation, since the initial wave influences were too weak.
**4.1 Sea Surface Temperature (SST) and 2-m Air Temperature (T02)**
Figure 3a and Figure 4a show the spatial distribution of 53-day (day 4 to day 56) averaged SST bias
in CTRL in boreal winter and summer, respectively. Here the bias in Fig.3a&4a is defined as SST in
CTRL minus OISST. To highlight the differences of the other four experiments (Table 1) versus the
CTRL, a percentage relative difference (PRD) of the bias is computed as $\text{PRD} = \frac{|\hat{y}_s - y| - |\hat{y}_c - y|}{|y|} \times 100\%,$
where $y$ is OISST, $\hat{y}_c$ is simulated SST in CTRL and $\hat{y}_s$ is simulated SST in other experiments (Fig.
3c-f and Fig. 4c-f). A negative value of PRD indicates that the bias is smaller compared to CTRL, and
vice versa.
In boreal winter, the global mean SST bias is approximately 0.30℃, and the average root mean square
error (RMSE) is about 0.90℃ from day 4 to day 56 in CTRL (Fig. 3a). The simulated SST is generally
overestimated, and the large biases (>1.0℃) are mainly distributed in the Southern Ocean. The regression
coefficients of absolute SST bias in CTRL (absolute value of bias in Fig. 3a) with time are almost positive
everywhere (Fig. S3a in the supplementary), indicating that the model biases increase with time. Figure





3b shows the distribution of correlation coefficients between the absolute biases with positive regression
coefficients in CTRL (absolute value of biases in Fig. 3a) and PRDs in ALL (Fig. 3f) from day 4 to day
56. Only coefficients with P value less than 0.05 are shown. The negative (positive) values indicate that
the SST bias in CTRL decreases (increases) with time in the fully coupled ALL experiment. Noticeably,
the bias reduction mainly occurs in regions where SST is largely overestimated (>1.0℃), particularly in
the Southern Ocean.
To understand the key process responsible for the bias reduction in ALL, the global distribution of
averaged PRD was compared across all four experiments (Fig.3c-f). Clearly, the distribution of PRD in
experiment VR12-AL-SC-EN is similar to that in ALL (Fig. 3c&3f). The spatial correlation coefficient
between the average PRD distributions of the two experiments (Fig. 3c&3f) is 0.99, and the RMSEs of
SST are not different significantly (red and yellow lines in Fig. 5a). The global mean PRD in ALL is -
7.74±2.21%, and in most areas the PRD are significant (P ≤0.05) (Fig. 3f). Large SST improvements
mainly appear in the Southern Ocean, with a regional mean PRD reaching -35.89±9.98% south of 45°S
(Fig. 3f and red line in Fig. 5a). The reduction of overestimated SST in CTRL (red in Fig 3a) is because
the Langmuir turbulence parameterization with Stokes–Coriolis force and entrainment injects turbulent
kinetic energy into the ocean, which enhances vertical mixing, and subsequently cools the surface waters
(Belcher et al., 2012; Li et al. 2016). The modified roughness and relative velocity in Z0-FAN and FLUX
also influence upper ocean mixing (Fig. 3d&e) via changing momentum flux, but the total SST changes
are not significant (purple and blue lines in Fig. 5a). The effect from Stokes drift-related ocean mixing
parameterizations dominate SST changes in ALL.
In boreal summer, the global mean SST bias in CTRL is overestimated approximately 0.29℃, and the
averaged RMSE from day 4 to day 56 is about 0.87℃. The overestimated SSTs (>1.0℃) mainly occur



in the Northern Hemisphere (Fig. 4a). The cooling effects in VR12-AL-SC-EN lead to a global mean
PRD of -3.85±1.32%, smaller than that in boreal winter, and the large SST improvements mainly occur
north of 50°N (Fig. 4c and yellow line in Fig. 5b). The changes of SST in Z0-FAN and FLUX (Fig. 4d,
e; purple and blue lines in Fig. 5b) are again relatively small. The global mean PRD in ALL is -3.26±
1.13% (Fig. 4f). The correlation coefficients between absolute bias in CTRL (Fig. 4a) and PRD in ALL
(Fig. 4f) show significant negative values mainly in the Northern Hemisphere (Fig. 4b), indicating
improvements of overestimated SST in ALL. While both positive and negative PRDs of SST appear in
the Southern Ocean, probably owing to the insufficient model resolution which could not resolve
mesoscale activities.

As aforementioned, large improvements of overestimated SST mainly occur in mid and high latitudes

in local summer. The time series of RMSEs and correlation coefficients of SST between model and
observation in the region (0-360°E, 45°-78°S in boreal winter and 0-360°E, 50°-78°N in boreal summer)
are shown in Fig. 5c-f. The RMSE in CTRL (blue in Fig. 5c&d) increases in the first few weeks and then
gradually decreases afterward. Compared with CTRL, RMSEs in VR12-AL-SC-EN (yellow) and ALL
(red) are significantly (P≤0.01) reduced by about 0.3℃. The spatial correlation coefficients decrease
with time but remain high (>0.90) for all experiments (Fig. 5e&f). The experiments ALL and VR12-AL-
SC-EN (red and yellow) show higher coefficients than CTRL (blue), indicating that the Stokes drift-
related ocean mixing plays a dominant role in SST warm bias reduction in high latitudes in summer.

We also compared T02 from experiments with ERA5 (Fig. 6). Warm biases of T02 appear in both

winter and summer in CTRL (Fig. 6a&b). The distribution of PRD of T02 is generally consistent with
that of SST (Fig.3&4), because the surface air temperature is mainly regulated by SST. In boreal winter,
all wave-coupled experiments reduce the T02 bias in general (Fig.6c-f). VR12-AL-SC-EC has the largest
T02 bias reduction. The global averaged PRD of bias is -7.51±0.11%, with reduction exceeding 20% in
the Southern Ocean (Fig.6c). In boreal summer, bias of T02 is slightly increased in the Southern Ocean
in all experiments (Fig.6g-j), consistent with the slight SST bias increase in the Southern Ocean (Fig. 4c-
f).
**4.2 Mixed Layer Depth (MLD)**
To further evaluate the direct effect of the wave-related processes on upper ocean, we compared the
MLD of all experiments with that estimated from Argo profiles in summer. The simulated T/S were
interpolated onto the positions of Argo profiles at the nearest time. The MLD was estimated as the depth
where the change of potential density reaches the value corresponding to a 0.2℃ decrease of potential
temperature with unchanged salinity from surface (de Boyer Montégut et al., 2004; Wang and Xu, 2018).
The time series of MLDs from numerical experiments and Argo south of 45°S in boreal winter (north
of 45°N in boreal summer) are compared in Fig. 7a (7b). The simulated MLDs are generally within the
STD of Argo MLDs (shading in Fig. 7). In CTRL, the mean bias (CTRL minus Argo) with STD is -
13.15±7.82 m (-6.75±5.29 m) in boreal winter (summer). The correlation coefficients of MLDs in CTRL
with Argo MLDs is 0.55 (0.68) with P ≤0.01, and the mean RMSEs is 15.30 m (8.55 m) in boreal winter
(summer). In ALL, the mean bias (ALL minus Argo) with STD is 9.61±8.78 m (5.23±7.22 m) in boreal
winter (summer). The correlation coefficient of MLDs south of 45°S (north of 45°N) enhances to 0.69
(0.79) in boreal winter (summer). The RMSE south of 45°S decreases from 15.30 m in CTRL to 13.02
m. The RMSE north of 45°N decreases from 6.71 m in CTRL to 4.93 m in ALL in the first six weeks but
the value increases in the last two weeks due to overestimation of MLD. Compared with CTRL (orange
in Fig. 7), VR12-AL-SC-EN (yellow) and ALL (dark blue) show significantly improvements (P ≤0.01)



on the underestimated MLD time series, whereas the MLD difference between CTRL and Z0-FAN
(purple)/FLUX (blue) is non-significant. Furthermore, there is no significant difference between VR12-
AL-SC-EN and ALL, indicating the Stokes drift-related ocean mixing dominates the total wave effects
on MLD in ALL.
**4.3 Wind Speed at 10 m (WSP10) and Significant Wave Height (SWH)**
In general, the WSP10s in CTRL are overestimated compared with ERA5 (Fig.8a and Fig. 9a). The
global averaged RMSEs of WSP10 are 3.51 m/s (3.53 m/s) in boreal winter (summer). The correlation
coefficients between absolute bias in CTRL and PRD in ALL are calculated where regression coefficients
of absolute bias with time are positive in CTRL (Fig. S3c&d). Only values, significant at 95% confidence
level, are shown. The widely-distributed negative values indicate decreasing trends of bias in these areas
(Fig. 8b&9b). In addition, the comparisons of WSP10 from numerical experiments with ERA5 (Fig.
8a&9a) indicate that the overestimated (underestimated) WSP10 corresponds to the overestimated
(underestimated) SWH (Fig. 10a&11a).
The comparisons of the simulated SWH with the ERA5 data over 53 days for boreal winter and boreal
summer are shown in Figure 10 and 11, respectively. In boreal winter, the global mean SWH bias in
CTRL is approximately 0.20 m, mainly due to the overestimates (> 0.30 m) in the Pacific, the North
Atlantic and the Southern Ocean (Fig.10a), and the average RMSE is about 1.04 m. In boreal summer,
the global mean bias in CTRL is approximately 0.17 m with 0.96 m RMSE (Fig. 11a). The significant
negative correlation (Fig. 10b&11b) between absolute bias in CTRL and PRD in ALL appears in most
overestimated regions, indicating the improvement of overestimated SWH in the fully coupled ALL
experiment.





WSP10 and SWH differences between sensitivity experiments and CTRL (sensitivity experiments
minus CTRL) are shown (Figs. 8-11 c-f). For boreal winter, in VR12-AL-SC-EN, the reduction of SST
warm bias affects air temperature and stabilizes marine atmospheric boundary layer (Sweet et al. 1981;
O'Neill et al. 2003), and subsequently reduces WSP10 and SWH with global averaged bias reduction of
-0.31 $\pm$ 0.14 m/s and -0.19±0.09 m (Fig.8c&10c), respectively. In Z0-FAN experiment, the
overestimated WSP10s and SWHs are also reduced, and the global averaged bias reduction is -0.30±0.14
m/s and -0.23±0.12 m (Fig. 8d&10d). The changes in WSP10 and SWH stem from the updated $z_0$,
generated by WW3 with the ST4-FAN scheme, which is larger than the original $z_0$ in GFS at wind
speed > 10 m/s (Fig. 2). The increase of $z_0$ enhances wind stress and momentum transferred into the
ocean, and therefore reduces surface winds (Pineau-Guillou et al. 2018; Sauvage et al. 2020). Whereas
the reduced surface winds decrease $z_0$ (Fig. 2), therefore in the end the wind stresses are weaker in most
areas in Z0-FAN than in CTRL. Since the 10-m wind in GFS is diagnosed from the friction velocity by
the well-known logarithmic profile (Charnock, 1955), its change is consistent with wind stress, and
consequently reduces SWH. In FLUX, the change of WSP10 and SWH is marginal in terms of global
average (Fig. 8e&10e). $\overrightarrow{U_{surf}}$ and $\overrightarrow{u_s(0)}$ in FLUX decrease wind stress and momentum transfer when
their directions are consistent with that of wind, and vice versa (Hersbach and Bidlot, 2008; Renault et
al., 2016). Although the decreased momentum flux into water could increase surface wind (Renault et
al., 2016), the effect is rather small in the coupled system with a relative coarse resolution. The effects
of $\overrightarrow{U_{surf}}$ and $\overrightarrow{u_s(0)}$ are discussed separately in Section 4.4. With all combined effects, biases in ALL
in most region tend to decrease (Fig. 8f&10f) compared with CTRL (Fig. 8a&10a), especially for
overestimated WSP10 and SWH.
In boreal summer, the changes of WSP10 and SWH are relatively small in terms of global average. In





ALL, the global averaged bias reduction of WSP10 (SWH) is -0.14±0.13 m/s (-0.16±0.07 m). The largest
reduction primarily appears in the Southern Ocean (Fig. 9f&11f) to improve the overestimated westerly
and SWH in CTRL (Fig. 9a&11a). Notably, the WSP10 and the SWH significantly increase in the south
of Australia and the northwestern Pacific in FLUX (Fig. 9e&11e), where they are underestimated in
CTRL. The WSP10 and the SWH are enhanced by the modified momentum transfer caused by the Stokes
drift and ocean surface current (Fig. S1d&f). Even though, the increases partly vanish in ALL (Fig.
9f&11f) owing to stabilization of atmosphere caused by Stokes drift-related ocean mixing (Fig. 9c&11c)
and enhanced roughness (Fig. 9d&11d).
Previous studies indicated that ocean surface winds in ERA5 are underestimated in some regions
(Belmonte Rivas and Stoffelen 2019; Kalverla et al. 2020; Sharmar and Markina 2020). To better
demonstrate the effects of wave on WSP10 and SWH, we also compared the simulation results with
NDBC buoy data (locations shown in Fig. 12, numbers of available buoys marked in Table 2 and buoy
identifiers listed in Table S4). According to the CTRL bias distribution, the biases (CTRL minus NDBC)
were sorted into three categories (Table 2), that is, the upper quartile (UQ; the buoys with the highest 25%
bias; yellow marks in Fig. 12), the lower quartile (LQ; the buoys with the lowest 25% bias; light blue
marks) and the median (MD; the rest 50%; black marks). And the corresponding mean biases in CTRL
were given in brackets of Table 2. In general, the UQ/LQ buoys (yellow/light blue in Fig.12) collocate
with positive/negative bias between CTRL and ERA5 (shading in Fig. 12), and the MD buoys collocate
with mild bias. From Table 2, the biases of both WSP10 and SWH between CTRL and buoy data are
positive in both seasons, except for LQ buoys.
The 53-day mean absolute percentage errors (MAPE=$(^{100\%}/_n) \sum_{i=1}^{n} \left| \frac{\hat{y}_i - y_i}{y_i} \right|$, where $\hat{y}_i$ is simulated
value, $y_i$ is NDBC buoy observation, i=1, 53) were calculated. The method from Hsu et al. (1994) was





used to adjust wind speeds from buoy data to the reference height of 10 m. The corresponding MAPE
differences compared with CTRL for the other four simulations are shown in Table 2, where a negative
(positive) value means that the bias is reduced (enhanced) versus CTRL.
In boreal winter, all four experiments lead to significant improvements on WSP10 and SWH for UQ
at 95% confidence level. The maximum MAPE difference for WSP10 is -17.31% in FLUX, and -11.93%
for SWH in Z0-FAN. While the MAPE differences for LQ are not significant, and the differences for
MD are small too. On average, the FLUX experiment shows the best improvement on both WSP10 and
SWH, which is -4.97% and -4.43% respectively.
In boreal summer, the MAPE improvements are larger than in boreal winter, and the differences are
lower than -10% for WSP10 for UQ in all experiments. This is consistent with the distribution of bias
improvement compared with ERA5 (Figs. 8-11). The largest MAPE difference is -23.21% for WSP10
in FLUX, and -20.05% for SWH in Z0-FAN. Again, the MAPE differences for LQ and MD are relatively
small. On average, the largest improvements are shown in FLUX and Z0-FAN, with the MAPE
difference of -7.51% and -6.99% for WSP10 and SWH, respectively. Therefore, the wave-related
processes are most effective in areas with large positive biases. The most effective process varies,
depending on seasons and locations.
**4.4 Enthalpy Fluxes and Effects of Stokes Drift on Air-Sea Fluxes**
The enthalpy fluxes in CTRL are shown in Fig. S4 of supplementary, which are positive upwards. The
latent heat flux differences between sensitivity experiments (Table 1) and CTRL are shown in Fig. 13.
Note that the distribution of the differences is consistent with those of WSP10 (Fig. 8&9) in general,
because the increase of $\Delta \vec{V}$ leads to enhanced upward latent heat flux, and vice versa. Meanwhile, the





differences in sensible heat flux between sensitivity experiments and CTRL have similar distribution
patterns but weaker magnitude (Fig. S5 of supplementary).
To better understand the effects of current and Stokes drift on air-sea fluxes in CFSv2, two extra
experiments with only surface current (FLUX_CURR) or Stokes drift (FLUX_ST) considered in Eqn.
10 were carried out. We compared the difference of momentum flux, latent heat flux and sensible heat
flux between FLUX_CURR and CTRL (Fig. S6 in supplementary), as well as FLUX_ST and CTRL (Fig.
S7 in supplementary), respectively. Since the effects of $\overrightarrow{U_{surf}}$ and $\overrightarrow{u_s(0)}$ on $\Delta \vec{V}$ depend on their
directions, enhanced (reduced) momentum flux and WSP10 occur in areas with relatively large (small)
angles with wind (Fig. S1g-j). The angles between Stokes drift and wind are much smaller than those
between current and wind, and the large angles in mid-high latitudes and coastal areas are resulted from
the dominance of swells. For instance, the northeasterlies over Kuroshio in boreal summer (Fig. S1b) is
in alignment with the Stokes drift (Fig. S1f), reducing the wind stress and the WSP10 (Fig. S7b, d). In
contrast, the Kuroshio is northeastward (Fig. S1d), enhancing wind stress and WSP10 (Fig. S6b, d).
Compared with the NDBC data (Fig. 12), the combined effects of the current and the Stokes drift lead
to larger MAPE improvements in FLUX than those in both FLUX_CURR and FLUX_ST for WSP10
(Table S3). Note that these effects are stronger in mid and high latitudes in winter due to large winds and
large waves. In addition, the latent heat flux as well as the sensible heat flux change consistently with
momentum flux (Fig. S6e-h&S7e-h).
**5 Summary and Discussion**
To investigate the individual role played by wave-related processes on atmosphere and ocean interface
in a coupled global atmosphere-ocean-wave modeling system on intraseasonal scale, we implemented



the version 5.16 of WW3 into CFSv2.0 for global oceans from 78ºS-78ºN, using the C-Coupler2. In this
coupled system, the WW3 was forced by 10-m wind generated in GFS. Stokes drift-related Langmuir
mixing, Stokes-Coriolis force and entrainment in ocean, air-sea fluxes modified by surface current and
Stokes drift, and momentum roughness length ($z_0$) were considered separately, and the results of
sensitivity experiments were compared against in-situ buoys, satellite measurements and ERA5
reanalysis. The effects of waves on intraseasonal prediction were examined in two 56-day cases, one for
boreal winter and the other one for boreal summer.
The following key results were found:
1.    Overestimated SST, T02 and underestimated MLD in mid and high latitudes in CFSv2.0 are

significantly improved, particularly in local summer. Because enhanced vertical mixing

generated by Langmuir turbulence, Stokes-Coriolis force and entrainment in VR12-AL-SC-

EN changes temperature structure in the upper ocean, and further affects air temperature. In

boreal winter, the global averaged SST (T02) is improved by 6.90±2.26% (7.51±0.11%).

Especially for the Southern Ocean, the regional mean improvement of SST reaches up to 35.89

±9.98%. In boreal summer, the effect is weaker because of the smaller ocean areas in mid and

high latitudes of the Northern Hemisphere, with a 3.85±1.32% global mean improvement in

SST.

2.    In general, all wave-related processes lead to reduction of biases for WSP10s and SWHs,

particularly in regions where WSP10s and SWHs are overestimated. The decreased SST in

VR12-AL-SC-EN stabilizes marine atmospheric boundary layer, and leads to weakened

WSP10s and SWHs. The modified roughness in Z0-FAN generally reduces momentum

transfer into the ocean, and so decreases WSP10s and SWHs. The relative wind-wave-current



speed in FLUX also affects wind stress, and further influences WSP10s and SWHs. Compared
with NDBC observations, for buoys with overestimations in CTRL, in boreal winter the
maximum improvement for WSP10 is 17.31% in FLUX, and 11.93% for SWH in Z0-FAN. In
boreal summer, the largest improvement is 23.21% for WSP10 in FLUX, and 20.05% for SWH
in Z0-FAN.
As shown in Fig. 3&4, SST biases also appear in tropical oceans. In the work of Breivik et al. (2015),
considered the surface waves, the bias of SST simulation in the tropics is reduced mainly due to drag
from swells. The similar effect was also shown in the one-way coupled Nucleus for European Modelling
of the Ocean (NEMO) model with the Météo-France wave model (MFWAM), where the momentum as
well as the energy flux across the air-sea interface are refined (Law-Chune and Aouf, 2018). Besides,
more wave-related processes should be considered, such as sea spray, wave breaking and non-breaking
wave effects (Bao et al. 2019; Couvelard et al. 2020). All these processes are worth future evaluation.
Different parameterizations for the same wave-related process also deserve discussion. For ocean
surface roughness, the most classic parametrizations are those developed by Janssen (1989, 1991), Taylor
and Yelland (2001) and Drennan et al. (2003). The method of Taylor and Yelland (2001) requires the
peak wavelength for the total spectrum, whereas that of Drennan et al. (2003) only requires the peak of
wind-sea waves. This difference leads to the fact that the former is more suitable for a mixed sea state,
while the latter is more suitable for a young sea state (Drennan et al., 2005). And the effect of Janssen's
parameterization (1989, 1991) is similar to that of Drennan et al. (2003), since it is also based on the
wind-sea conditions (Shimura et al., 2017). In addition, Janssen's formulation can be modified to account
for the decreased drag coefficient for strong winds (Bidlot et al. 2020; ECMWF 2020), and the
parameterization can improve simulation of tropical cyclone (Li et al. 2021).





The case studies indicate that there remain significant biases in the coupled system, probably owing
to inaccuracy of coarse resolution, absence of a coupled wave-ice modular, and deficiency of initial fields.
In addition, to further improve the model and eliminate the biases, as Breivik et al. (2015) proposed,
extra adjusting of the individual model components in the coupled systems is also necessary. All of these
require further efforts to investigate efficient methods to improve fully coupled systems.
**Code and data availability**
The code developed for the coupled system can be found under https://doi.org/10.5281/zenodo.5109521
(Shi et al., 2020), including the coupling, preprocessing, run control and postprocessing scripts. The
initial fields for CFS are generated by the real time operational Climate Data Assimilation System,
downloaded from the CFS official website (http://nomads.ncep.noaa.gov/pub/data/nccf/com/cfs/prod).
The daily average satellite Optimum Interpolation SST (OISST) data are obtained from NOAA
(https://www.ncdc.noaa.gov/oisst), and the National Data Buoy Center (NDBC) buoy data are also
obtained from NOAA (https://www.ndbc.noaa.gov). The Argo observational profiles of T/S are available
at China Argo Real-time Data Center (www.argo.org.cn). The ERA5 reanalysis are available at the
Copernicus    Climate    Change    Service    (C3S)    Climate    Date    Store
(https://cds.climate.copernicus.eu/cdsapp#!/dataset/reanalysis-era5-single-levels).
**Author contribution**
FX and RS designed the experiments and RS carried them out. RS developed the code of coupling
parametrizations and produced the figures. ZF contributed to the installation and operation of CFSv2.0.
LL and HY contributed to the application of C-Coupler2. XL and YZ provided the original code of



CFSv2.0. RS prepared the manuscript with contributions from all co-authors. FX and HL contributed to
review and editing.
**Acknowledgments**
The authors would like to extend thanks to all developers of the CFSv2.0 model
(https://cfs.ncep.noaa.gov/cfsv2/downloads.html). The work was supported by National Key Research
and Development Program of China (no. 2016YFC1401408), and Tsinghua University Initiative
Scientific Research Program (2019Z07L01001).

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





**Table 1**. List of numerical experiments: setups different from CTRL are marked with bold

| Experiments | Physical Process/Parameterization | | |
| --- | --- | --- | --- |
| | Langmuir Cell with Stokes-Coriolis Force and Entrainment | Roughness (Charnock Parameter) | Relative Velocity in Flux |
| CTRL | Off | Off | Off |
| VR12-AL-SC-EN | **Eqn. 1-3, 5, 6** | Off | Off |
| Z0-FAN | Off | $C_{ch}$ **from Eqn. 12, 13** | Off |
| FLUX | Off | Off | $\Delta\vec{V}$ **from Eqn. 10** |
| ALL | **Eqn. 1-3, 5, 6** | $C_{ch}$ **from Eqn. 12, 13** | $\Delta\vec{V}$ **from Eqn. 10** |



**Table 2.** The 53-day mean absolute percentage error (MAPE) differences for SWH and WSP10 between VR12-AL-SC-EN/ Z0-FAN/ FLUX/ ALL and CTRL (MAPE in VR12-AL-SC-EN/ Z0-FAN/ FLUX/ ALL minus MAPE in CTRL); the MAPE is calculated as MAPE=$\left(\frac{100\%}{n}\right)\sum_{i=1}^{n}\left|\frac{\hat{y}_i-y_i}{y_i}\right|$, where $\hat{y}_i$ is simulated value, $y_i$ is NDBC buoy observation, i=1, 53; UQ, LQ and MD mean the 25% buoys with the highest bias in CTRL, the 25% buoys with the lowest bias in CTRL and the rest 50%; values in brackets are the mean bias in CTRL (CTRL minus NDBC); the asterisk means statistically significant of the difference compared with CTRL at 95% confidence level.

| WSP10 MAPE Difference (20170106-20170228) for 374 buoys | | | |
|---|---|---|---|
| **CTRL Bias (m/s)** | **UQ**CTRL (3.28) | **LQ**CTRL (-1.15) | **MD**CTRL (0.67) | **TOTAL** (0.87) |
| VR12-AL-SC-EN | -6.46* | 1.52 | 1.16* | -0.67 |
| Z0-FAN | -9.99* | 5.37 | 0.90* | -0.71* |
| FLUX | -17.31* | -0.05 | -1.22* | -4.97* |
| ALL | -11.78* | 3.85 | -0.04* | -2.02* |

| SWH MAPE Difference (20170106-20170228) for 175 buoys | | | |
|---|---|---|---|
| **CTRL Bias (m)** | (0.73) | (-0.23) | (0.14) | (0.20) |
| VR12-AL-SC-EN | -6.96* | -2.32 | 2.45* | -1.12* |
| Z0-FAN | -11.93* | 3.13* | 3.40* | -0.52* |
| FLUX | -9.06* | -5.64* | -1.48* | -4.43* |
| ALL | -11.25* | -1.90 | 3.25* | -1.69* |

| WSP10 MAPE Difference (20180806-20180928) for 404 buoys | | | |
|---|---|---|---|
| **CTRL Bias (m/s)** | (2.43) | (-1.52) | (0.28) | (0.37) |
| VR12-AL-SC-EN | -13.52* | 1.23 | -2.79* | -4.47* |
| Z0-FAN | -15.75* | 1.63 | -0.19* | -3.14* |
| FLUX | -23.21* | 1.25* | -4.04* | -7.51* |
| ALL | -22.53* | 0.50 | -3.65* | -7.33* |

| SWH MAPE Difference (20180806-20180928) for 181 buoys | | | |
|---|---|---|---|
| **CTRL Bias (m)** | (0.58) | (-0.04) | (0.29) | (0.28) |
| VR12-AL-SC-EN | -5.64* | 11.08 | 2.58* | 2.65* |
| Z0-FAN | -20.05* | 5.05* | -6.48* | -6.99* |
| FLUX | -5.25* | 0.17 | -3.64* | -3.09* |
| ALL | -13.50* | 4.63* | -2.77* | -3.60* |





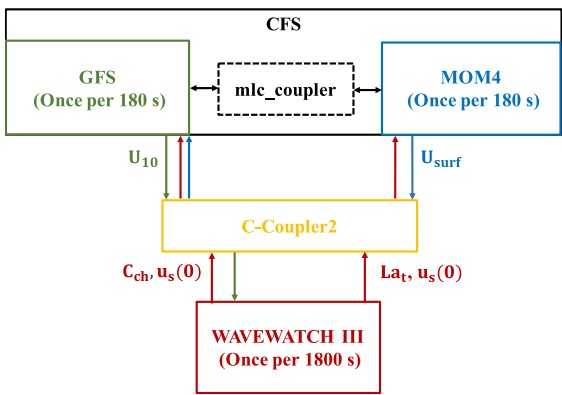

**Figure 1.** A schematic diagram of the atmosphere-ocean-wave coupled modeling system. The arrows indicate the coupled variables that are passed between the model components. In the diagram, $C_{ch}$, $La_t$, $u_s(0)$, $U_{10}$, and $U_{surf}$ are Charnock parameter, turbulent Langmuir number, surface Stokes drift velocity, 10-m wind and surface current, respectively.





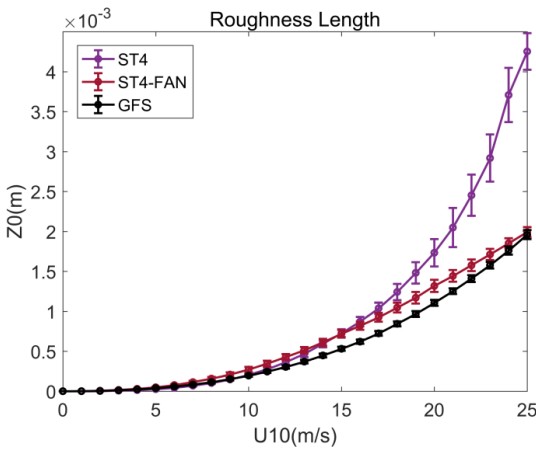

**Figure 2.** Relationships between momentum roughness length $z_0$ (m) in the coupled system and 10-m wind speed

(m/s); error bars indicate twice the standard deviations for each point.



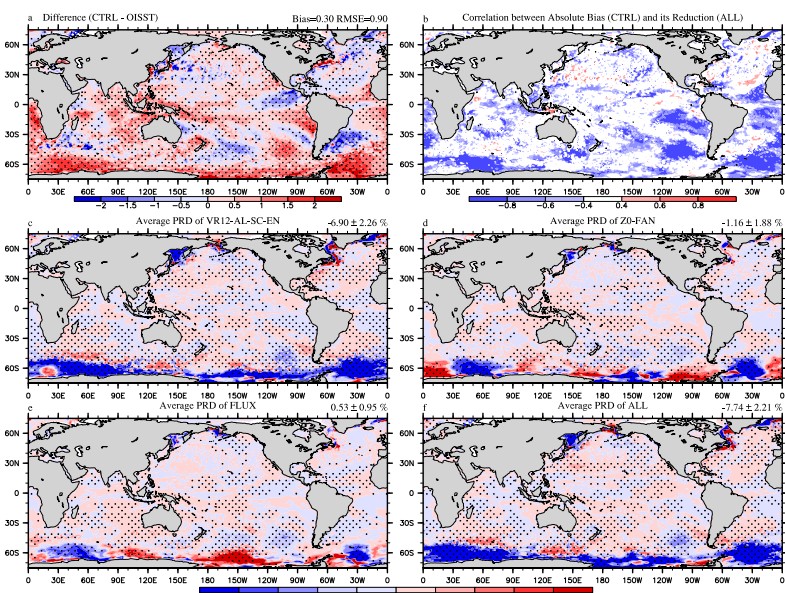

**Figure 3.** The 53-day average SST (℃) bias in CTRL, the correlation between absolute CTRL bias and its relative reduction, and percentage relative difference of bias in Jan-Feb, 2017: **a** the SST bias between CTRL and OISST (CTRL minus OISST), **b** the correlation between absolute value of CTRL bias and percentage relative difference in ALL (only values corresponding to statistically significant at 95% confidence level and positive regression coefficient of absolute bias in CTRL are shown), **c/d/e/f** the percentage relative difference between VR12-AL-SC-EN/Z0-FAN/FLUX/ALL and CTRL. The percentage relative difference is a percentage computed as $\mathrm{PRD} = \frac{|\hat{y}_s - y| - |\hat{y}_c - y|}{|y|} \times 100\%$, where $y$ is OISST, $\hat{y}_c$ is simulated SST in CTRL and $\hat{y}_s$ is simulated SST in other experiments, so a negative value means that the error is smaller than that of CTRL, and vice versa; dotted areas are statistically significant at 95% confidence level.





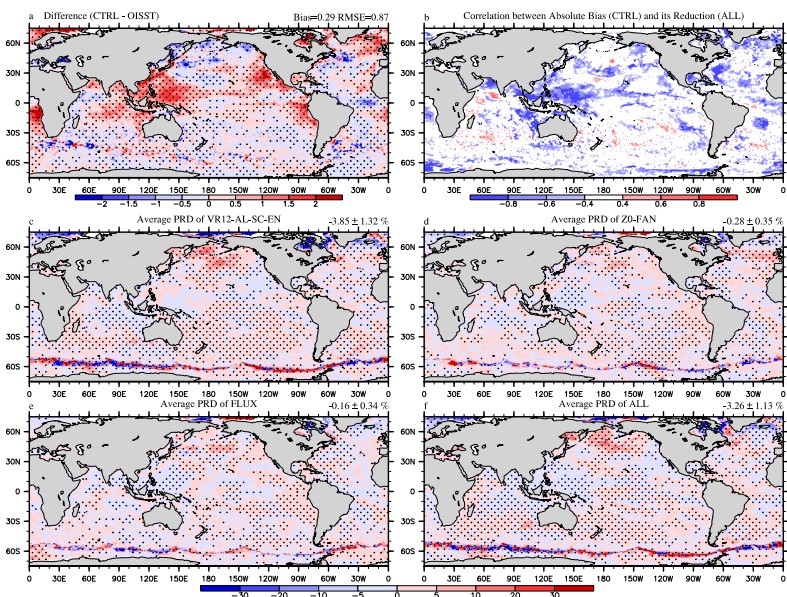

**Figure 4.** As Fig. 3, but for the 53-day average SST (℃) bias in CTRL, the correlation between absolute CTRL bias

and its relative reduction, and percentage relative difference of bias in Aug-Sep, 2018.



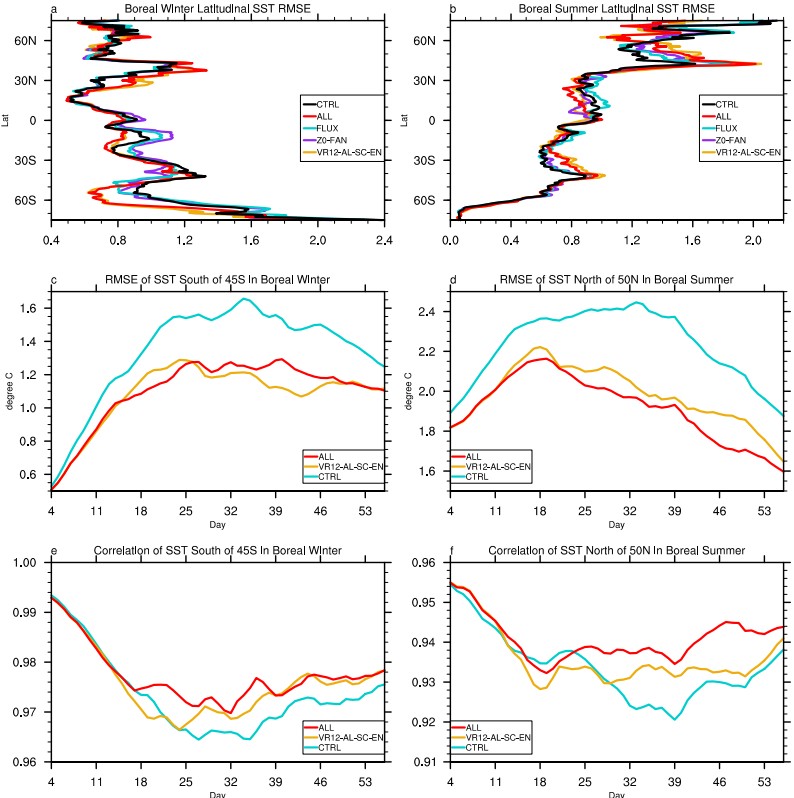

**Figure 5.** The 53-day averaged latitudinal distribution of SST root mean square errors (RMSE), time series of domain-averaged SST RMSE and correlation coefficient: **a/b** the latitudinal RMSE in boreal winter/summer compared with OISST, **c/d** the time series of domain-averaged (0-360°E, 45-78°S/50-78°N) SST RMSE in boreal winter/summer, **e/f** the time series of domain-averaged (0-360°E, 45-78°S/50-78°N) SST correlation coefficient in boreal winter/summer; differences of RMSE and correlation coefficient time series between VR12-AL-SC-EN/ALL and CTRL are statistically significant at 99% confidence level.





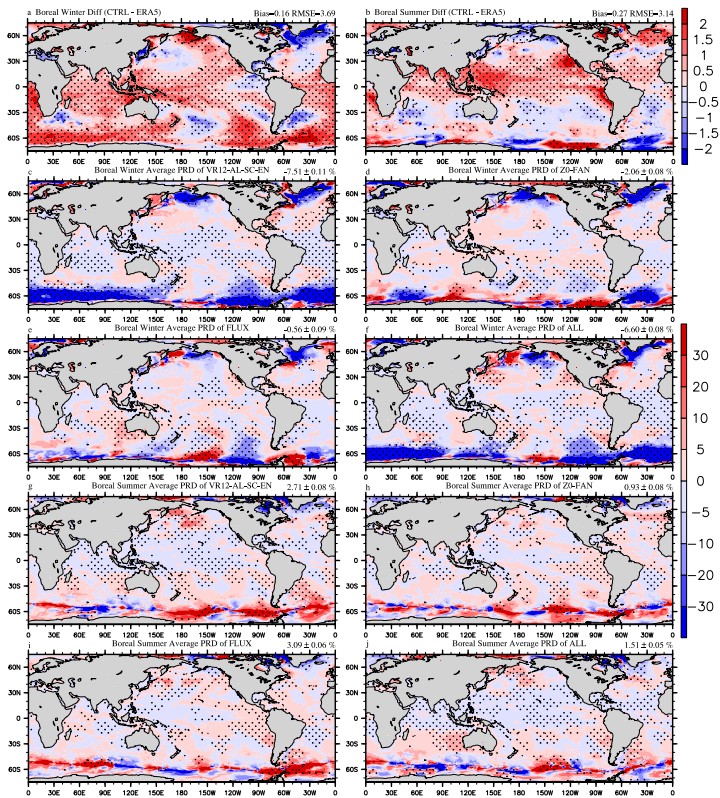

**Figure 6.** The 53-day average T02 (℃) bias and its percentage relative difference in Jan-Feb, 2017 and Aug-Sep, 2018: **a/b** the T02 bias between CTRL and ERA5 (CTRL minus ERA5) in boreal winter and summer; **c/d/e/f** the percentage relative difference between VR12-AL-SC-EN/Z0-FAN/FLUX/ALL and CTRL in Jan-Feb, 2017; **g/h/i/j** the same as c/d/e/f but for Aug-Sep, 2018. The percentage relative difference is a percentage computed as PRD = $\frac{|\hat{y}_s - y| - |\hat{y}_c - y|}{|y|} \times 100\%$, where $y$ is ERA5, $\hat{y}_c$ is simulated T02 in CTRL and $\hat{y}_s$ is simulated T02 in other experiments, so a negative value means that the error is smaller than that of CTRL, and vice versa; dotted areas are statistically significant at 95% confidence level.



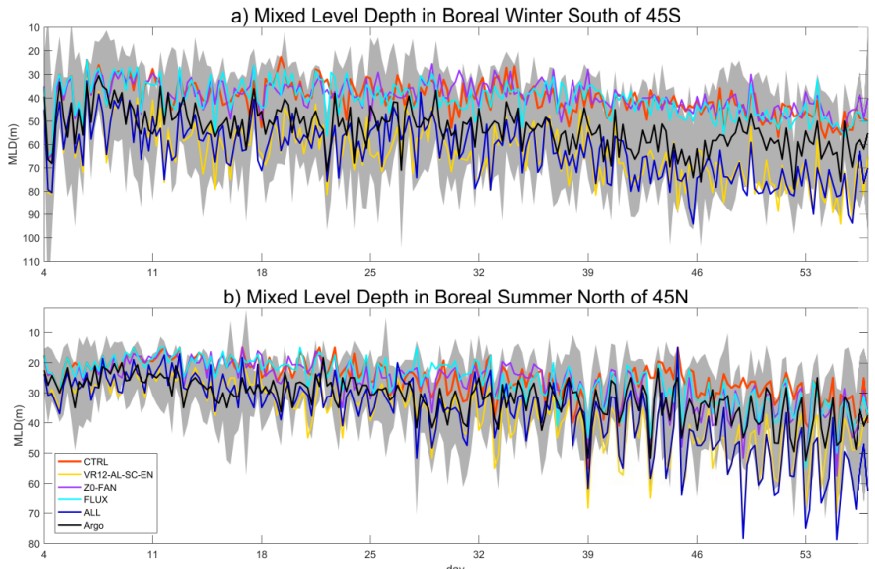

**Figure 7.** The 53-day time series of domain-averaged (0-360°E, 45-78°S/N) mixed layer depth (MLD; m) in boreal

winter/summer: the difference between CTRL and VR12-AL-SC-EN/ALL passes the student's t-test at 99%

confidence level; the time intervals are 6 hours; shaded areas indicate twice the standard deviations for Argo.



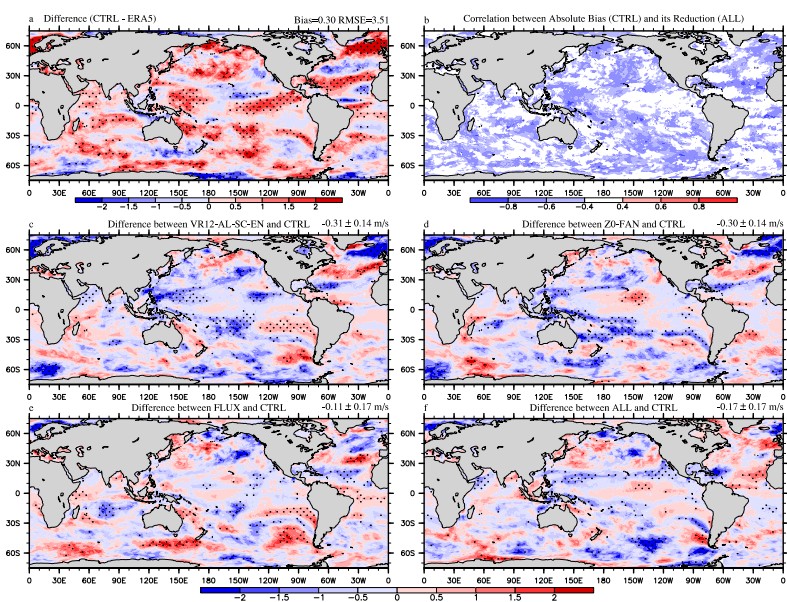

**Figure 8.** The 53-day average WSP10 (m/s) bias in CTRL, the correlation between absolute CTRL bias and its relative reduction, and the difference between 4 sensitivity experiments and CTRL in Jan-Feb, 2017: **a** the WSP10 bias between CTRL and ERA5 (CTRL minus ERA5), **b** the correlation between absolute value of CTRL bias and percentage relative difference in ALL (only values corresponding to statistically significant at 95% confidence level and positive regression coefficient of absolute bias in CTRL are shown), **c/d/e/f** the difference between VR12-AL-SC-EN/Z0-FAN/FLUX/ALL and CTRL (VR12-AL-SC-EN/Z0-FAN/FLUX/ALL minus CTRL); dotted areas are statistically significant at 95% confidence level.



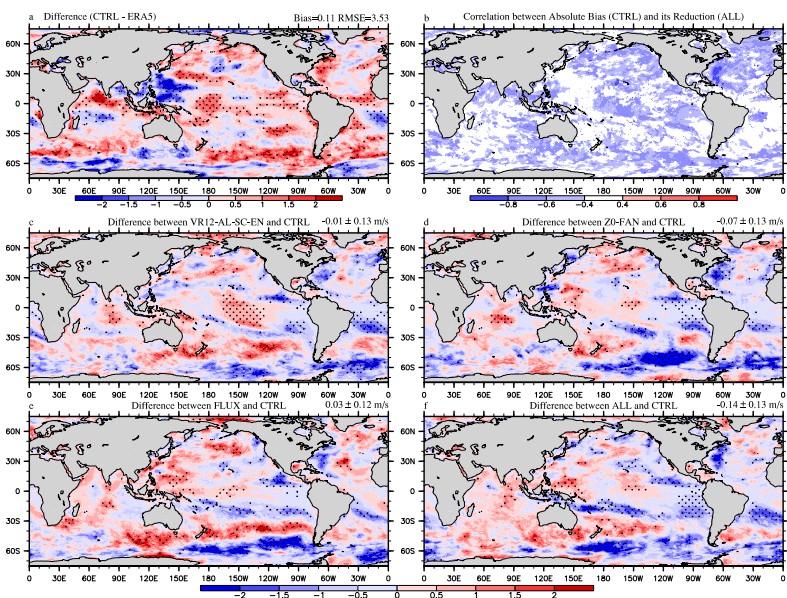

**Figure 9.** As Fig. 8, but for 53-day average WSP10 (m/s) bias in CTRL, the correlation between absolute CTRL

bias and its relative reduction, and the difference between 4 sensitivity experiments and CTRL in Aug-Sep, 2018.



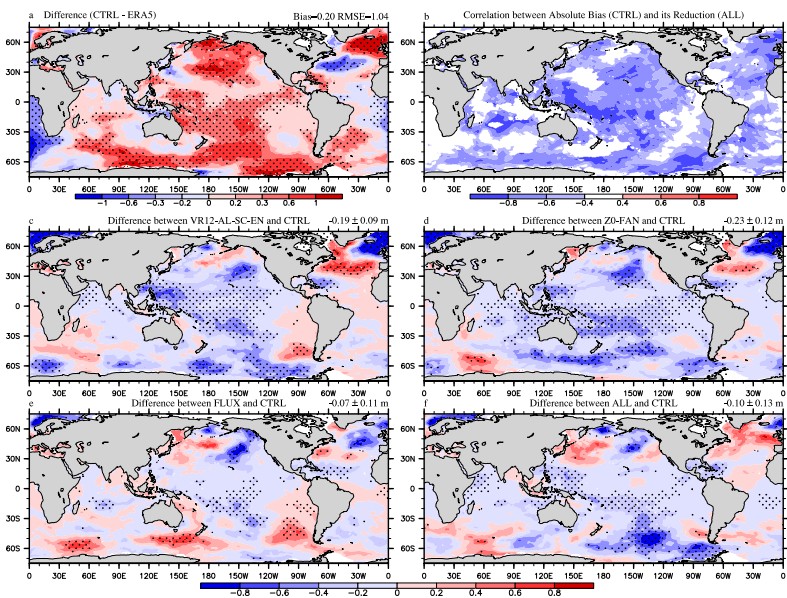

**Figure 10.** The 53-day average SWH (m) bias in CTRL, the correlation between absolute CTRL bias and its relative reduction, and the difference between 4 sensitivity experiments and CTRL in Jan-Feb, 2017: **a** the SWH bias between CTRL and ERA5 (CTRL minus ERA5), **b** the correlation between absolute value of CTRL bias and percentage relative difference in ALL (only values corresponding to statistically significant at 95% confidence level and positive regression coefficient of absolute bias in CTRL are shown), **c/d/e/f** the difference between VR12-AL-SC-EN/Z0-FAN/FLUX/ALL and CTRL (VR12-AL-SC-EN/Z0-FAN/FLUX/ALL minus CTRL); dotted areas are statistically significant at 95% confidence level.



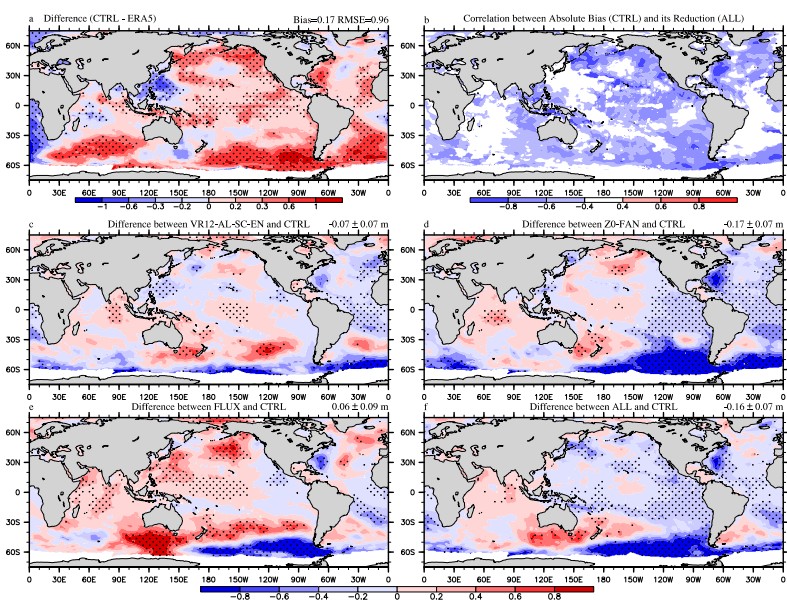

**Figure 11.** As Fig. 10, but for the 53-day average SWH (m) bias in CTRL, the correlation between absolute CTRL bias and its relative reduction, and the difference between 4 sensitivity experiments and CTRL in Aug-Sep, 2018.





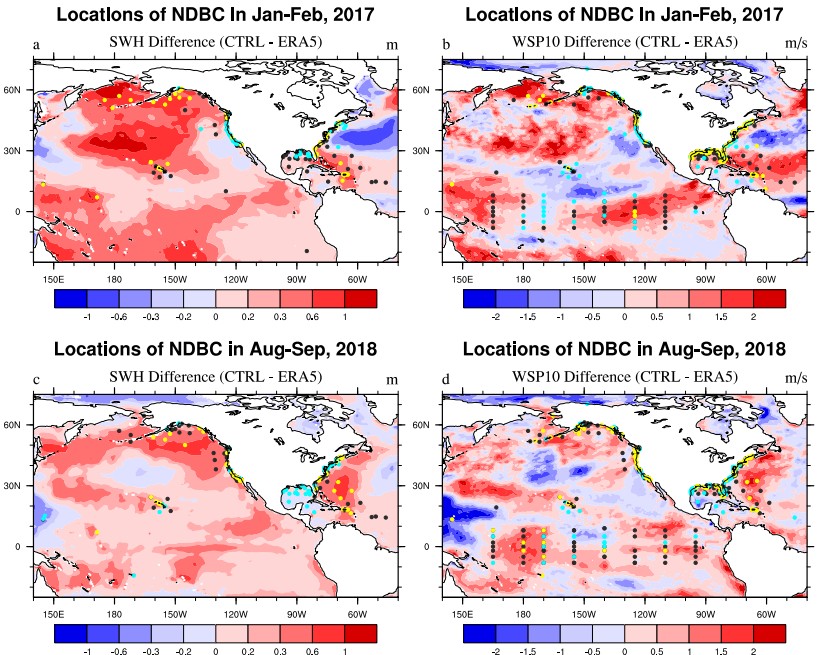

**Figure 12.** The locations of NDBC buoy data; shaded areas are 53-day averaged SWH biases (**a&c**) and WSP10 biases (**b&d**) between CTRL and ERA5 (CTRL minus ERA5); yellow, light blue and black marks are the 25% buoys with the highest bias in CTRL (UQ; CTRL minus NDBC), the 25% buoys with the lowest bias in CTRL (LQ) and the rest 50% (MD).



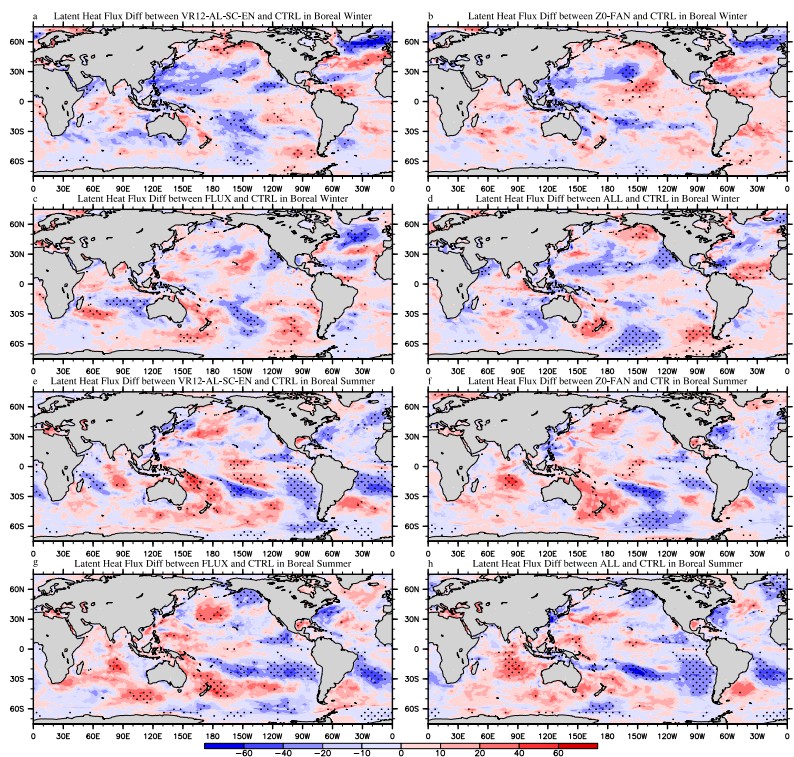

**Figure 13.** The 53-day average latent heat flux difference (W/m2) between 4 sensitivity experiments and CTRL in Jan-Feb, 2017 and Aug-Sep, 2018; a-d: the difference between VR12-AL-SC-EN/Z0-FAN/FLUX/ALL and CTRL (VR12-AL-SC-EN/Z0-FAN/FLUX/ALL minus CTRL) in Jan-Feb, 2017; e-h: the difference between VR12-AL-SC-EN/Z0-FAN/FLUX/ALL and CTRL in Aug-Sep, 2018; the enthalpy fluxes are positive upwards; dotted areas are statistically significant at 95% confidence level.