# Peer review of "The Effects of Ocean Surface Waves on Global Intraseasonal Prediction: Case Studies with a Coupled CFSv2.0-WW3"

_Geoscientific Model Development, 2021_

## Author Comment (AC1)

**Review from Referee #1**

*This paper is a resubmission of a paper I already made comments about as a reviewer (https://doi.org/10.5194/gmd-2020-327). I thank the authors for re submitting an updated version of their work, and congratulate them for the general improvement of the manuscript, and for taking into account most of my comments. Especially, I appreciate that they use longer time spans for the sensitivity tests. The statistical analysis of the results is also much improved.*

*The present paper is a case study investigating the effects of several parameterizations representing the impact of waves on the ocean surface layer (Langmuir mixing and Stokes-Coriolis force with entrainment) and atmosphere surface layer (change of roughness length, effect of surface currents on the turbulent fluxes). The CFS2.0 ocean-atmosphere climate model and the WAVEWATCHIII wave model (WW3) are used in coupled mode for global simulations at resolution 0.25° to 0.5° for two time periods of 53 days, in boreal summer and winter. Four different simulations enable to assess the different effects on the SST, ocean mixed layer depth (MLD), 10-m wind speed, significant wave height (SWH) and latent heat flux in an incremental way. The conclusion is that refining the CFS2.0 representation of the surface exchanges by including additional terms due to waves leads to an overall (although modest) improvement of the SST and MLD biases with respect to observations. The improvement is larger in the Southern Ocean in boreal winter. Some improvement is also obtained on the surface wind speed and SWH, compared to ERA5.*

*The results presented here are not especially new, as recent sensitivity studies using the same kind of modeling platforms showed similar effects (e.g. Shimura et al., 2017; Torres et al., 2018; Bao et al., 2020; Couvelard et al., 2020;). But the sensitivity of the system CFS2.0-WW3 to these wave effects has not been studied so far.*

*Nevertheless, I have several major comments about the description of the coupled system, the evaluation of the impact of the different parameterizations, and the interpretation of the results.*

Response: We sincerely appreciate the reviewer for her/his constructive comments on the manuscript. Our responses are listed as follows in blue. Text are revised accordingly.

*General comments*

*1- Part 2 describes the representation of several physical effects impact the wave-ocean or wave-atmosphere exchanges, which has been implemented in the coupled system. The effects of Stokes-Coriolis and Langmuir mixing come as additional terms in the Richardson number or turbulent velocity scale of the KPP mixing scheme, the wave effect on the atmospheric roughness length comes through a change of the Charnock parameter, and the effect of the surface currents corresponds to the use of the relative surface velocity in computing the turbulent fluxes. For the effect of the Langmuir mixing, the authors assume that wind and waves are aligned, arguing the effect of misalignment has been shown to be non significant by Li et al. 2016. However, other studies like Polonichko (1997), Van Roekel et al. (2012), and Li et al (2017) showed that the Langmuir cell intensity strongly depends on the alignment between the Stokes drift and wind direction. The latter study especially concluded that assuming alignment of wind and waves leads to excessive mixing, particularly in winter. As the strongest effect of the Stokes-Coriolis and Langmuir mixing parameterization is obtained on the Southern Ocean in winter, I suggest to mention the results of*

*these works in comparing the results of the VR12-AL-SC-EN experiment with respect to the CTRL one. Also Couvelard et al (2020) showed that there is a significant difference between annual averages of the module of the surface Stokes drift and of the part that is aligned with the wind (their Fig. 2). Please discuss.*

*Also, the description of the exchanges of the different parameters between model compartments is unclear to me. I understand that all additional terms are computed in WW3, and that the Stokes drift and Langmuir mixing terms are transferred to MOM4, that the Stokes drift is transferred to GFS for computed the surface relative wind, and that the Charnock parameter is also transferred to GFS for computing the surface roughness (Fig.1). What is unclear is what is exchanged between GFS and MOM4? Especially, are the (regular) surface currents transferred from MOM4 to GFS and used for estimating a relative wind velocity in computing the turbulent fluxes by GFS? If so, is it consistent with the transfer and use of the Stokes drift from WW3? Please provide the corresponding information, with an update of Fig.1. What is the meaning of the blue arrow from the coupler to GFS in Fig.1?*

*About the effect of the surface current on the atmosphere: I guess from eq. 7 to 9 and section 2.4 that only the effect of the currents (and especially of the Stokes drift) on the turbulent fluxes is taken into account, and not the effect of the current on the surface wind through the tridiagonal matrix (see the work of Lemarié 2015). If so, the fact that the coupling is not complete should be clearly stated in section 2.4.*

Response: We agree that the misalignment between the Stokes drift and wind direction is important for the intensity of Langmuir cell (Polonichko, 1997; Van Roekel et al., 2012; Li et al., 2017), particularly in the Southern Ocean (Couvelard et al., 2020). In the comparison of the VR12-AL-SC-EN experiment with respect to the CTRL, we add "Since we didn't consider the misalignment between the Stokes drift and wind direction for the Langmuir mixing, which is important in the Southern Ocean (Couvelard et al., 2020), the effects of the Stokes drift might be overestimated". Without the misalignment, however, the simulation still underestimated the mixing, and so we didn't consider misalignment in the study.

To clarify the exchanges between model components, we updated the original Fig. 1 as shown in Fig. R1. Between GFS and MOM4, originally GFS receives SST from MOM4, and sends fluxes of heat, momentum, and freshwater to MOM4 via mlc_coupler. In the study, the (regular) surface currents from MOM4 are transferred to GFS via C_Coupler2 to estimate the relative wind velocity for the turbulent fluxes (Eqn. 7-10) in GFS. Similarly, the Stokes drift from WW3 is also transferred to GFS. The blue arrows in Fig. R1 indicate the surface currents transferred from MOM4 to GFS and WW3. The text and Fig. 1 are revised accordingly.

To complete the coupling, we add the surface current and Stokes drift to the tridiagonal matrix (Lemarié 2015) in CFS for the FLUX experiment. The experiments are re-run and the associated figures and text are revised.

[Figure]

**Figure R1.** A schematic diagram of the atmosphere-ocean-wave coupled modeling system. The arrows indicate the coupled variables that are passed between the model components. In the diagram, $C_{ch}$, $La_t$, $u_s(0)$, tus, $U_{10}$, and $U_{surf}$ are Charnock parameter (red arrows), turbulent Langmuir number (red arrows), surface Stokes drift velocity (red arrows), Stokes drift transport (red arrows), 10-m wind (green arrows) and surface current (blue arrows), respectively.

*2- The statistical analysis of the different sensitivity tests is much clearer and easier to understand than in the previous version of the paper. I still feel rather uncomfortable with the different diagnostics used by the authors. For instance, the correlation between the bias reduction and the absolute bias shown in Fig. 3,4, 8 to 11 is almost never commented, and I am not sure about its meaning: from the text, I guess that its corresponds to the correlation between the relative change between CTRL and ALL (the so-called PRD) and the absolute bias, but only when the time evolution of the bias corresponds to an increase. Is it so? What is the additional information with respect to the PRD as shown elsewhere? Please elaborate.*

*For most of the parameters compared in this study, the maps represent the relative improvement (PRD). For the 10-m wind speed and MLD however, differences with the CTRL are given and I find these maps easier to read. Please justify why you use different diagnostics or homogenize. The relative improvement (PRD) depends strongly on the initial value of the bias. Why not showing maps of the biases for the different simulations? It would help to appreciate where the biases have been corrected or not. Please give the values of the final biases (and RMSE) for every parameter/experiment, in addition to the PRD.*

*The comparison of the 10-m wind speed and SWH with the NDBC follows some of my previous recommendation, and I thank the authors for that. I think, however, that the way this comparison is presented could be greatly improved. I suggested that maybe, the wind speed can influence the bias and the difference between CTRL and ALL, and this comment is still valid. There is some effect of the value of the bias with CTRL, even though the current presentation of the results makes it difficult to apprehend. Rather than a table giving the relative difference for different quantiles of biases, I would suggest using a graph comparing directly the 10-m wind speed of the simulation outputs (y-axis) with the 10-m wind speed of the NDBC buoys (x-axis) in wintertime (same in summertime, and for the SWH), every dot on the graph representing a buoy (4 graphs in total). The results of the different simulations can be plotted in the same graph, with different colors. This would enable a direct comparison, including the effect of the wind speed (x-axis) and of the bias (distance to the y=x line). The changes between the different simulations can be given by the mean biases and*

*standard deviations with respect to observations, rather than the relative mean changes.*

Response: The correlation between the absolute CTRL biases (CTRL minus OISST/ERA5) and PRD in ALL is replaced by the correlation between the absolute CTRL biases and the differences of absolute biases between ALL and CTRL ($|\widehat{y_a} - y| - |\widehat{y_c} - y|$, where y is OISST or ERA5, $\widehat{y_c}$ ($\widehat{y_a}$) is simulated result in CTRL (ALL)). In the updated figures (e.g. Fig.R2b-R4b for boreal winter), all values with statistically significant at 95% confidence level are shown. Note that the absolute bias of CTRL increases with time in most areas (Fig. S3 in the supplementary), where the negative (positive) correlation values indicate that the simulated bias in CTRL decreases (increases) with time in ALL experiment. While in areas where the absolute bias of CTRL decreases with time, the situation is reversed.

To clarify, we have replaced all maps of PRD with the differences between sensitivity experiments and CTRL. The global averaged biases and RMSEs are shown in the upper right of each map (e.g. Fig.R2-R4 for boreal winter). The figures for boreal summer are also updated. The text is revised accordingly.

As suggested, the comparisons of the 10-m wind speed and SWH with the NDBC buoys are shown in Fig. R5. The x-axis is WSP10/SWH of buoys, and the y-axis is the simulated WSP10/SWH in Jan-Feb, 2017 (Fig. R5a&b) and Aug-Sep, 2018 (Fig. R5c&d) for all experiments. The mean biases with standard deviations and RMSEs for every experiment are shown in Table R1. The differences between 4 sensitivity experiments and CTRL are all statistically significant at 95% confidence level. In CTRL, the WSP10/SWH is generally overestimated in both winter and summer with positive mean bias (Table R1). Except the FLUX experiment in Jan-Feb, 2017, all other experiments lead to a decrease of mean bias by reducing WSP10 and SWH. With the increase of WSP10, the reduction of overestimation is more obvious. In ALL experiment, the RMSEs decrease compared with CTRL.

[Figure]

**Figure R2.** The 53-day average SST (°C) bias in CTRL (a), the time correlation between the absolute CTRL biases and the differences of absolute biases between ALL and CTRL (b), and the differences between VR12-AL-SC-EN (c)/Z0-M04 (d)/ FLUX (e)/ ALL (f) and CTRL in Jan-Feb,

2017. The first 3-day simulation is discarded. The dotted areas are statistically significant at 95% confidence level.

[Figure]

**Figure R3.** The same as Figure R2 but for WSP10 (m/s).

[Figure]

**Figure R4.** The same as Figure R2 but for SWH (m).

[Figure]

**Figure R5.** Scatter plots of simulated WSP10/SWH (y-axis) vs buoy WSP10/SWH (x-axis): (a) the WSP10 in Jan-Feb, 2017, (b) the SWH in Jan-Feb, 2017, (c) the WSP10 in Aug-Sep, 2018, and (d) the SWH in Aug-Sep, 2018. The dotted line is y=x.

**Table R1.** The 53-day mean bias with standard deviation (STD) and RMSE for WSP10 and SWH compared with NDBC buoy observation: the bias is calculated as simulation minus NDBC.

| Boreal Winter WSP10 | Bias with STD | RMSE |
|:---:|:---:|:---:|
| CTRL | 0.15±1.27 | 1.28 |
| VR12-AL-SC-EN | 0.02±1.18 | 1.18 |
| Z0-M04 | -0.01±1.13 | 1.13 |
| FLUX | 0.39±1.25 | 1.30 |
| ALL | 0.05±1.11 | 1.11 |
| **Boreal Winter SWH** | **Bias with STD** | **RMSE** |
| CTRL | 0.21±0.38 | 0.44 |
| VR12-AL-SC-EN | 0.14±0.35 | 0.37 |
| Z0-M04 | 0.10±0.30 | 0.32 |
| FLUX | 0.24±0.34 | 0.42 |
| ALL | 0.12±0.34 | 0.36 |
| **Boreal Summer WSP10** | **Bias with STD** | **RMSE** |

| | Bias with STD | RMSE |
|---|---|---|
| **CTRL** | 0.17±1.26 | 1.27 |
| **VR12-AL-SC-EN** | 0.06±1.33 | 1.33 |
| **Z0-M04** | 0.02±1.24 | 1.24 |
| **FLUX** | -0.12±1.22 | 1.22 |
| **ALL** | -0.10±1.18 | 1.18 |
| **Boreal Summer SWH** | **Bias with STD** | **RMSE** |
| **CTRL** | 0.28±0.25 | 0.38 |
| **VR12-AL-SC-EN** | 0.20±0.24 | 0.31 |
| **Z0-M04** | 0.22±0.26 | 0.34 |
| **FLUX** | 0.14±0.25 | 0.29 |
| **ALL** | 0.12±0.21 | 0.24 |

*3- Interpreting the results could, again, be made in a more accurate and concise way. For instance, the discussion in section 4.3 is rather long and not very easy to follow. Probably the effects in boreal summer with respect to boreal winter could be presented more briefly.*

*Overall, I am not sure commenting in details improvements of a few percent is meaningful, but adding information about the correlation between the changes of different parameters can help to interpret the results. For instance, what is the correlation between the 2-m temperature and the SST changes? It confirm that the SST change is actually at the origin of the 2-m temperature change. Also, the correlation between the bias changes in 10-m wind speed and SWH is probably high. Please give values and discuss.*

*At some parts, the interpretation is not complete. For instance, the part about the latent heat flux does not lead to clear, concise results.*

*From Fig. S3, I understand that the time evolution of the absolute value of the biases of the different parameters considered is overall positive, both in winter and in summer. At most places, these trends are significant, and even large, like more than 0.02°C/day (corresponding to more than 1°C difference for the 53-day simulated period) or 0.02 m/day for SWH (more than 1 m difference). What is the implication for the mean biases, and their changes from CTRL to ALL? Does it mean that the simulations are drifting in time from their initial state, because there is no assimilation of data? Or, conversely, that their stationarity is not reached yet because the simulated period are too close from the initial state, despite the hot start? Please comment on that.*

*I specifically asked about the possible effects of including the parameterization of processes related with waves on the turbulent heat fluxes. The authors added a section about that effect, and I thank them for investigating it. However, it appears not significant, at least not for the time scales considered in this study.*

Response: We will modify the Section 4, especially the Section 4.3, to make the discussion concise. As suggested, the correlation coefficients between the SST and the T02 changes (ALL minus CTRL) in boreal winter and summer are calculated. The values are 0.60 and 0.51 respectively, significant at 99% confidence level, indicating the SST change is actually the origin of the 2-m temperature change. The correlation coefficients between WSP10 difference and SWH difference in ALL compared with CTRL are 0.77 and 0.73 in boreal winter and summer, significant at 99% confidence level, indicating the SWH change is originated from wind speed. The text is revised accordingly.

For Figure S3 in CTRL, we think the simulations are drifting in time from the initial state due to no data assimilation, and the effects in ALL partially limit this. In Figure 5 of the main text, the RMSE

of SST increases in the first twenty days and then decreases with time. So the stationarity has been reached in the simulation.

The turbulent heat fluxes change is primarily resulted from the change of 10-m wind speed, and it is non-significant due to the relative short simulation period. So the related discussion is removed in the text.

*Detailed comments*

*l. 26-29: are you sure that the SST change is at the origin of the 10-m wind speed change? I understand from 4.3 that the change of z0 also plays a role. Please check.*

Response: We apologize for the confusion. The corresponding lines are changed as "The largest regional mean SST improvement occurs in the Southern Ocean. For WSP10 and SWH, the wave-related processes generally lead to reduction of biases in regions where wind speed and SWH are overestimated. The decreased SST stabilizes marine atmospheric boundary layer, weakens wind speed and then SWH. The increased roughness length due to waves leads to reduction in the originally overestimated wind speed and SWH. Meanwhile, the effects of Stokes drift and current on air-sea fluxes also play a role in change of WSP10 and SWH."

*l. 65-66: the studies cited here are at climate scale, not for numerical prediction. The only model including wave effects and used for numerical prediction is ECMWF (IFS-WAM).*

Response: We agree. The text is revised to "The overall effects of wave-related processes have been shown to be important in global coupled systems for both climate research (e.g., Law-Chune and Aouf, 2018; Bao et al. 2019; Couvelard et al. 2020) and numerical prediction (Breivik et al. 2015)".

*l. 112-114: this set of experiment follows one of my previous question (is there any impact of the coupling frequency?). Table S1 brings some statistics but they are not commented in the text. Please justify why 1800s was chosen as the coupling frequency.*

Response: The coupling frequency indeed influences the simulation results slightly. To quantify, the RMSEs of SST, SWH and WSP10 with different coupling steps for the ALL experiment are shown in Table R2. From Table R2, the 10_STEP_WW3 experiment has the closest RMSEs to the 1_STEP_ALL for SST, SWH and WSP10, and has the relatively small runtime. Therefore, the time steps of 10_STEP_WW3 are selected to compromise computer time consumption and the model RMSE. Note that the original one time step in CFS is 180 s, and 10 time steps are 1800 s. Text is revised to clarify.

**Table R2.** The 28-day global RMSEs and daily runtime for SST, SWH and WSP10 in the ALL experiment with different coupling steps in Jan, 2017. In 1_STEP_ALL experiment, three model components are coupled every time step, and in 5_STEP_ALL (10_STEP_ALL) they are coupled every 5 (10) steps. Particularly, in 10_STEP_WW3, only the WW3 is coupled every 10 time steps, whereas the GFS and the MOM4 remain the one time step coupling frequency as the original settings in CFS.

| Experiments | SST (℃) | SWH (m) | WSP10 (m/s) | Daily runtime (s) |
|---|---|---|---|---|
| 1_STEP_ALL | 0.88 | 1.09 | 3.94 | 25677 |
| 5_STEP_ALL | 0.88 | 1.09 | 3.88 | 19546 |

| 10_STEP_ALL | 0.88 | 1.23 | 4.25 | 19012 |
| 10_STEP_WW3 | 0.88 | 1.10 | 3.95 | 19171 |

*l. 122-124: "the daily initial fields at 00:00 UTC.." I guess it is rather the initial field of the first day of each experiment? Or is the model re initialized every 24h from the operational analysis? Please specify.*

Response: Yes, the initial field here refers to the first day of each experiment, not re-initialized every 24h from the operational analysis. The text is revised to "the initial fields at 00:00 of the first day in each experiment".

*l. 285 and following: Couvelard et al (2020) also obtained improvement of SST/MLD biases in the Southern Ocean. Please discuss your results against theirs.*

Response: Based on the coupling of NEMO and WW3, Couvelard et al. (2020) considered the wave-supported stress, the modification of Charnock parameter, the Stokes drift-related forces, the Langmuir cell and the modified turbulence kinetic energy, and the SST and MLD biases were largely improved. The main differences between our study and Couvelard et al. (2020) are three aspects: (1) Couvelard et al. (2020) considered the misalignment of wind and waves for Langmuir cells, which leads to weaker mixing than assuming alignment; (2) the modification of wave-supported stress and turbulence kinetic energy; (3) the initial SSTs are generally underestimated in Couvelard et al. (2020; Fig. 13a&14a), especially in the Southern Ocean. While in our system, we don't have the second parametrizations, and even the application of VR12-AL (assuming alignment of wind and waves) cannot produce strong enough mixing in the Southern Ocean, and subsequently cannot large reduce the warm bias of SST and the shallow bias of MLD. Therefore, we further include the effects of Stokes–Coriolis force and entrainment (VR12-AL-SC-EN) to enhance mixing. Our results also show that the VR12-AL-SC-EN experiment leads to reasonable improvement of SST and MLD, especially in the Southern Ocean in boreal winter. Text is revised to clarify the differences between our results and theirs.

*l. 315-316: "generally consistent", please quantify.*

Response: To quantify, the spatial correlation coefficients between the SST and the T02 change in ALL relative to CTRL are calculated, which are 0.60 and 0.51 in boreal winter and summer respectively, significant at 99% confidence level.

*l. 332 and following: please give the correlation coefficient between ALL MLD and the observations, so they can be compared with the 0.55 value given for CTRL.*

Response: The correlation coefficient of MLDs south of 45°S (north of 45°N) with Argo observations in CTRL is 0.55 (0.68), while the correlation coefficient of MLDs in ALL enhances to 0.69 (0.79). Text is revised accordingly.

*l. 348: I do not know how to interpret the "negative trends of bias". What is the meaning of that?*

Response: We apologize for the confusion. We updated Figure 8b&9b. The "negative trends of bias" is deleted.

*l. 349-351: I would rather say the opposite: the biases in SWH are directly related to the biases of*

*the 10-m wind speed.*
Response: Text is revised as suggested.

*l. 390 and following: see general comments. A graph showing the biases (model vs obs) in the different simulations would be easier to understand. Also, listing the buoy numbers in the SI is probably not useful, especially without additional information (position, number of observations). Please indicate, for every comparison, the number of buoys used.*
Response: As suggested, we replace Table 2 with Fig. R5 and Table R1. The number of buoys used for every figure and corresponding buoy identifiers with longitude and latitude are listed in the revised supplementary.

*Section 4.4: investigating the heat exchanges is nice at climate scale, but probably not relevant at the time scale of 2 months. I asked to authors to check for that, but it seems that the latent heat flux is directly influenced by the 10-m wind. Plus, the discussion in this part does not lead to clear results (to me). What would be your conclusion, beyond "the latent heat flux depends on the wind speed only"?*
Response: Yes, we only conclude that the latent heat flux change depends on the wind speed. Since the wave-related effect on the turbulent heat fluxes is non-significant for the 2-month simulation, we remove the Section 4.4.

References

Bao, Y., Song, Z., & Qiao, F. (2020). FIOâ☐☐ESM version 2.0: Model description and evaluation. Journal of Geophysical Research: Oceans, 125, e2019JC016036. https://doi.org/ 10.1029/2019JC016036

Couvelard, X., Lemarié, F., Samson, G., Redelsperger, J. L., Ardhuin, F., Benshila, R., & Madec, G. (2020). Development of a two-way-coupled ocean–wave model: assessment on a global NEMO (v3. 6)–WW3 (v6. 02) coupled configuration.GMD, 13(7), 3067-3090.

LemarieÌ☐, F., 2015: Numerical modification of atmospheric models to include the feedback of oceanic currents on air–sea fluxes in ocean–atmosphere coupled models. INRIA Grenoble-Rhône-Alpes Tech. Rep. RT-464, 10 pp. [Available online at https://hal.inria.fr/hal-01184711/document.]

Li, Q., Webb, A., Fox-Kemper, B., Craig, A., Danabasoglu, G., Large, W. G., & Vertenstein, M. (2016). Langmuir mixing effects on global climate: WAVEWATCH III in CESM. Ocean Modelling, 103, 145-160.

Li, Q., Fox-Kemper, B., Breivik, Ø., & Webb, A. (2017). Statistical models of global Langmuir mixing.Ocean Modelling, 113, 95-114.

Polonichko, V.: Generation of Langmuir circulation for nonaligned wind stress and the Stokes drift, J. Geophys. Res., 102, 15773– 15780, https://doi.org/10.1029/97JC00460, 1997.

Shimura, T., N. Mori, T. Takemi, and R. Mizuta (2017), Long-term impacts of ocean wave-dependent roughness on global climate systems, J. Geophys. Res. Oceans, 122, 1995–2011, doi:10.1002/2016JC012621.

Torres, O., Braconnot, P., Marti, O., & Gential, L. (2019). Impact of air-sea drag coefficient for latent heat flux on large scale climate in coupled and atmosphere stand-alone simulations. Climate Dynamics, (3), 2125-2144.

Van Roekel, L. P., Fox-Kemper, B., Sullivan, P. P., Hamlington, P. E., and Haney, S. R.: The form and orientation of Langmuir cells for misaligned winds and waves, J. Geophys. Res., 117, C05001, https://doi.org/10.1029/2011JC007516, 2012.

---

## Author Comment (AC2)

We sincerely appreciate the reviewer for her/his constructive comments on the manuscript. Our responses are listed as follows in blue. Text are revised accordingly.

**Review from Referee #2**

*Here are my comments.*
*Lines 150, 152 and 157 Van -> Van Roekel*
Response: As suggested, the text is revised.

*Line 169, section 2.3.2: the Stokes drift should also be used in the advection of any tracer, including temperature and also in the calculation of the vertical velocity in difference/convergence term NEMO4.*
Response: We agree. According to the work of Couvelard et al. (2020), we calculate the Stokes drift profile and add it to the corresponding advection terms and convergence terms in MOM4. The text is revised, and the results are updated.

*Line 179, section 2.4: there is an inconsistency in considering the impact of the surface current and of the surface Stokes drift on the momentum flux in the atmosphere model, but not in the wave model. With ST4 (and ST3), the surface momentum balance is re-evaluated in order to determine the friction velocity that is then used as part of the source terms calculation, and hence the evolution of the wave field. To be consistent, WW3 should be forced not with the absolute 10m wind, with the relative 10m wind with respect to the surface current.*
Response: As suggested, the relative 10-m wind with respect to the surface current is applied in the wave model as well. Figure R1 and text are updated accordingly.

[Figure]

**Figure R1.** A schematic diagram of the atmosphere-ocean-wave coupled modeling system. The arrows indicate the coupled variables that are passed between the model components. In the diagram, $C_{ch}$, $La_t$, $u_s(0)$, tus, $U_{10}$, and $U_{surf}$ are Charnock parameter (red arrows), turbulent Langmuir number (red arrows), surface Stokes drift velocity (red arrows), Stokes drift transport (red arrows), 10-m wind (green arrows) and surface current (blue arrows), respectively.

*Line 216: as far as I can understand from the text, the ST4-Fan scheme is used for z0 in the atmospheric model only, and not in WW3. This is not consistent and should be made clearer that z0*

*and hence u\* inside WW3 will still be based on a Charnock determined from a modified version of Janssen wave induced stress (Ardhuin et al. 2010).*

Response: To clarify, we revise Section 2.5.2 to address that the new z0 scheme (previously ST4-Fan and now revised to ST4-M04) is used in CFS, while the z0 in WW3 is calculated by the ST4 source term (Ardhuin et al., 2010) with the method of wave-induced kinematic stress (Janssen 1989, 1991).

*Also, WW3 can be run with a cap on z0, and hence on the Charnock it could return to the atmosphere. See z0max in table 2.6 in the WW3 manual. It is indeed set to a large value for TEST473f, however, TEST500 has z0max=0.002 for instance. I believe, it is worth mentioning as ST4-Fan is not the only way to limit Charnock for high winds. One should also mention some recent developments on modifying ST4 for high winds (Bidlot et al. 2020 and Li et al. 2021), without a very awkward parameterization of Fan et al. (sorry I notice that it is mentioned later (line 488), but it might too late).*

Response: As suggested, the introduction of recent developments on modifying ST4 for high winds (Bidlot et al. 2020 and Li et al. 2021) is moved from Section 5 to Section 2.5.2.

*I have serious problem with ST4-Fan. It will indeed limit Charnock for large winds, but from figure S2, it does not seem to make sense that Charnock is largest in the Tropics. The Charnock parameter was introduced to represent the impact of waves on the momentum transfer at the sea surface. It was recognized that young wind-sea should extract more momentum than older more mature old sea. So why is Charnock largest in the Tropics where the sea state should be dominated by old wind-sea and swell?*

Response: We agree that the calculation of Charnock parameter in ST4-Fan is problematic for the sea state dominated by old wind-sea and swell. To solve this problem, we revise ST4-Fan based on Moon et al. (2004). According to Fan et al. (2012), the equations of Charnock parameter are derived based on the observations in Moon et al. (2004). In Moon et al. (2004), the Charnock parameter decreases with the increase of wave age at low-middle winds (<30 m/s), but levels off or increases at high winds (>30 m/s; Fig. 3 of Moon et al. 2004). Moon et al. (2004) proposed Eqn. R1 to estimate the Charnock parameter, and gave different values of $a$ and $b$ every 5 m/s ranging from 10 m/s to 50 m/s (Table 1 of Moon et al. 2004). Based on this, Fan et al. (2012) proposed Eqn. R2 to calculate $a$ and $b$. Because $b$ is always positive (Eqn. R2), the Charnock from Eqn. R1 increases with wave age even at low wind speed, which generates large Charnock in the tropics.

$$C_{ch} = a(\frac{c_{pi}}{u_*})^b, \tag{R1}$$

$$a = \frac{0.023}{1.0568^{U_{10}}}, b = 0.012U_{10}, \tag{R2}$$

Therefore, we have re-derived the relationship (Eqn. R3) between $a/b$ and 10-m wind speed $U_{10}$ by fitting the values in Table 1 of Moon et al. (2004) for $U_{10}$>15 m/s,

$$a = \frac{1}{0.1477U_{10}^2 - 0.7395U_{10} - 10.9995},$$
$$b = 1.5661E^{-5}U_{10}^3 - 0.002U_{10}^2 + 0.1017U_{10} - 1.6182. \tag{R3}$$

Because the observations in Moon et al. (2004) are obtained under tropical cyclones, there is no reliable data for wind $\leq$ 15 m/s. In Eqn. R3, $b$ is negative (positive) from relatively small (large)

wind speed. So Eqn. R3 is used for $U_{10}>15$ m/s, whereas the original Charnock relationship of WW3 ST4 scheme (Janssen 1989, 1991) is used for $U_{10} \leq 15$ m/s. The revised parameterization is called ST4-M04. The comparison of $z_0$ among ST4-M04, ST4-Fan, ST4 and GFS is shown in Fig. R2, indicating that the new ST4-M04 is close to ST4-Fan at high wind speed and can improve $z_0$ at low wind speed. Consequently, the Charnock parameter in ST4-M04 is low in the Tropics as expected (Fig. R3).

[Figure]

**Figure R2.** Relationships between momentum roughness length $z_0$ (m) in the coupled system and 10-m wind speed (m/s); error bars indicate twice the standard deviations for each point.

[Figure]

**Figure R3.** The Charnock parameter $C_{ch}$ obtained by ST4-M04 in boreal winter (a) and summer (b).

*Line 231: it is more than just the fetch, but also the development stage of the sea state (young sea, old sea…)*

Response: As suggested, the text is revised to "Since the z0 is determined only by wind-sea conditions in ST4 and ST4-M04 scheme, the STD at a given wind speed is owing to variations in wind fetch and development stage of sea state. The reduced STDs in ST4-M04 scheme, compared to ST4, imply less sensitivity of z0 to fetch and sea state".

*Line 233, before section 2.6: Could you say what is done in WW3 when sea ice is present. It seems that there is a large impact near or within the sea ice, so it is important to know how the different fields were produced by WW3. Quite often, WW3 is set-up so that the wave spectra are reset to 0 every time step for all areas with a sea ice cover above a certain threshold. If it is the case, this will mean that the estimate for Stokes drift and wave age would be those for very young windsea. It can then be debated whether these estimates are correct, as it is known that in the presence of sea ice, the high frequency waves are heavily damped.*

Response: In WW3, the sea ice concentrations are from CFSR data (Saha et al., 2014). The ice blocking IC0 source term with the critical ice concentration of 50% is applied. So for ice concentration in the range of 0-50%, the wave spectra are not zero, while for ice concentration larger than 50% the wave spectra are zero. Indeed, the estimate for Stokes drift and wave age would be those for young wind-sea, inconsistent with the phenomenon that high-frequency waves are severely

damped in the presence of sea ice. To diminish the influence, since the estimate of roughness and fluxes in the presence of sea ice are different with those in open oceans in CFS, we turn off the coupling from WW3 to CFS to avoid any conflicts with sea ice. Therefore, the effect of wave-state in these areas on air-sea flux is not considered. Text is revised to clarify.

*Line 256 and table S4: NDBC website reports its own buoy data, as well as buoy data from other NOAA agencies, such as the data from the TAO array, and a few other buoy data providers along the US. But it also reports surface observations from many coastal stations (i.e. on land, piers or towers) that are definitely not buoy. They also report observations from the oil and gas industry in the Gulf of Mexico. In table S4, anything that does not have a 5-digit identifier is probably not a buoy, and the oil and gas data from the Gulf of Mexico have an identifier like 423xx, 428xx. All those non buoy data should not be use in this analysis.*

Response: As suggested, all non-buoy data are removed. The text is revised accordingly.

*Line 261: section 4.1: I am surprised by the lack of impact of the FLUX experiment on the SST, in particular in the equatorial Pacific. My experience is that including the winds relative the surface current in the momentum flux (surface stress) used to force the ocean model has quite an impact on the SST around the equatorial Pacific. To be sure, could you confirm, as indicated that MOM4 is indeed forced with the surface stress as shown in equation (7).*

Response: We checked the code as suggested, and the MOM4 is indeed forced with the surface stress as Eqn. 7 in the text. Previously, we showed the SST changes by a percentage relative differences (PRD $= \frac{|\hat{y}_s - y| - |\hat{y}_c - y|}{|y|} \times 100\%$, where $y$ is OISST, $\hat{y}_c$ is simulated SST in CTRL and $\hat{y}_s$ is simulated SST in other experiments) in Fig. 3&4. Since the absolute value of OISST on the denominator is relatively large, the PRD looks low in lower latitude. In the revised text, all the maps of PRD are replaced by SST differences between sensitive experiments and CTRL (Fig. R4b&d). The SST differences between FLUX and CTRL around the equatorial Pacific are manifest.

[Figure]

**Figure R4.** The 53-day average SST (°C) bias in CTRL and the SST difference between FLUX and CTRL: (a) the SST bias between CTRL and OISST (CTRL minus OISST) in Jan-Feb, 2017, (b) the difference between FLUX and CTRL (FLUX minus CTRL) in Jan-Feb, 2017, (c) the SST bias

between CTRL and OISST (CTRL minus OISST) in Aug-Sep, 2018, and (d) the difference between FLUX and CTRL (FLUX minus CTRL) in Aug-Sep, 2018. Dotted areas are statistically significant at 95% confidence level.

*Line 474: it not true, the SST bias in the Tropics was not reduced due to drag from swell. Breivik et al. (2015) considered the impact of sea state dependent Charnock, using the formulation from Janssen on the air-side surface stress, which was modulated further considering the momentum flux balance between wind input and wave breaking to determine the ocean-side stress that is then used to force the ocean. Moreover, the impact of wave breaking was also considered as input to the upper ocean mixing scheme (TKE). All these effects had an impact on the SST. What was not discussed in Breivik et al. (2015) is the impact of surface currents on the SST response in the Tropical Pacific as that effect has already been introduced in the ECMWF system years before. As I mentioned earlier, I am surprised by the lack of sensitivity on the CFS system to surface currents in the Tropics.*

Response: As suggested, the text is revised to "In Breivik et al. (2015), considered the surface waves effects including wave-related Charnock parameter, modification of ocean-side stress, wave dissipation-related turbulent kinetic energy flux, Stokes-Coriolis force and Langmuir mixing, the bias of SST simulation in the tropics is reduced". Figure R4 shows that surface current and Stokes drift can influence SST in tropics. The values can be up to more than 0.5℃, comparable with the effect of modified ocean-side stress in Breivik et al. (2015; Fig.1). Text is revised to clarify the effects of surface current and Stokes drift on SST.

References
Ardhuin, F., Rogers, E., Babanin, A. V., Filipot, J., Magne, R., Roland, A., Der Westhuysen, A. V., Queffeulou, P., Lefevre, J. M., and Aouf, L.: Semiempirical Dissipation Source Functions for Ocean Waves. Part I: Definition, Calibration, and Validation, Journal of Physical Oceanography, 40, 1917-1941, http://dx.doi.org/10.1175/2010JPO4324.1, 2010.

Breivik, O., Mogensen, K., Bidlot, J., Balmaseda, M., and Janssen, P. A. E. M.: Surface wave effects in the NEMO ocean model: Forced and coupled experiments, Journal of Geophysical Research, 120, 2973-2992, http://dx.doi.org/10.1002/2014JC010565, 2015.

Couvelard, X., Lemarié, F., Samson, G., Redelsperger, J.-L., Ardhuin, F., Benshila, R., and Madec, G.: Development of a two-way-coupled ocean-wave model: assessment on a global NEMO(v3.6)-WW3(v6.02) coupled configuration, Geosci. Model Dev., 13, 3067-3090, https://doi.org/10.5194/gmd-13-3067-2020, 2020.

Fan, Y., Lin, S., Held, I. M., Yu, Z., and Tolman, H. L.: Global Ocean Surface Wave Simulation Using a Coupled Atmosphere–Wave Model, Journal of Climate, 25, 6233-6252, http://dx.doi.org/10.1175/JCLI-D-11-00621.1, 2012.

Janssen, P. A. E. M.: The Quasi-linear theory of wind wave generation applied to wave forecasting. J. Phys. Oceanogr. 21, 1631-1642. https://doi.org/10.1175/1520-0485(1991)021<1631:QLTOWW>2.0.CO;2, 1991.

Janssen, P. A. E. M.: Wave-induced stress and the drag of air flow over sea waves. J. Phys. Oceanogr. 19(6), 745-754. https://doi.org/10.1175/1520-1310 0485(1989)019<0745:WISATD>2.0.CO;2, 1989.

Moon, I., Ginis, I., and Hara, T.: Effect of surface waves on Charnock coefficient under tropical cyclones, Geophysical Research Letters, 31, http://dx.doi.org/10.1029/2004GL020988, 2004.

Saha, S., Moorthi, S., Wu, X., Wang, J., Nadiga, S., Tripp, P., Behringer, D., Hou, Y., Chuang, H., and Iredell, M. D.: The NCEP Climate Forecast System Version 2, Journal of Climate, 27, 2185-2208, http://dx.doi.org/10.1175/JCLI-D-12-00823.1, 2014.

---

## Author Response (AR1)

**Review from Referee #1**

*This paper is a resubmission of a paper I already made comments about as a reviewer (https://doi.org/10.5194/gmd-2020-327). I thank the authors for re submitting an updated version of their work, and congratulate them for the general improvement of the manuscript, and for taking into account most of my comments. Especially, I appreciate that they use longer time spans for the sensitivity tests. The statistical analysis of the results is also much improved.*

*The present paper is a case study investigating the effects of several parameterizations representing the impact of waves on the ocean surface layer (Langmuir mixing and Stokes-Coriolis force with entrainment) and atmosphere surface layer (change of roughness length, effect of surface currents on the turbulent fluxes). The CFS2.0 ocean-atmosphere climate model and the WAVEWATCHIII wave model (WW3) are used in coupled mode for global simulations at resolution 0.25° to 0.5° for two time periods of 53 days, in boreal summer and winter. Four different simulations enable to assess the different effects on the SST, ocean mixed layer depth (MLD), 10-m wind speed, significant wave height (SWH) and latent heat flux in an incremental way. The conclusion is that refining the CFS2.0 representation of the surface exchanges by including additional terms due to waves leads to an overall (although modest) improvement of the SST and MLD biases with respect to observations. The improvement is larger in the Southern Ocean in boreal winter. Some improvement is also obtained on the surface wind speed and SWH, compared to ERA5.*

*The results presented here are not especially new, as recent sensitivity studies using the same kind of modeling platforms showed similar effects (e.g. Shimura et al., 2017; Torres et al., 2018; Bao et al., 2020; Couvelard et al., 2020;). But the sensitivity of the system CFS2.0-WW3 to these wave effects has not been studied so far.*

*Nevertheless, I have several major comments about the description of the coupled system, the evaluation of the impact of the different parameterizations, and the interpretation of the results.*

Response: We sincerely appreciate the reviewer for her/his constructive comments on the manuscript. Our responses are listed as follows in blue. Text are revised accordingly.

*General comments*

*1- Part 2 describes the representation of several physical effects impact the wave-ocean or wave-atmosphere exchanges, which has been implemented in the coupled system. The effects of Stokes-Coriolis and Langmuir mixing come as additional terms in the Richardson number or turbulent velocity scale of the KPP mixing scheme, the wave effect on the atmospheric roughness length comes through a change of the Charnock parameter, and the effect of the surface currents corresponds to the use of the relative surface velocity in computing the turbulent fluxes. For the effect of the Langmuir mixing, the authors assume that wind and waves are aligned, arguing the effect of misalignment has been shown to be non significant by Li et al. 2016. However, other studies like Polonichko (1997), Van Roekel et al. (2012), and Li et al (2017) showed that the Langmuir cell intensity strongly depends on the alignment between the Stokes drift and wind direction. The latter study especially concluded that assuming alignment of wind and waves leads to excessive mixing, particularly in winter. As the strongest effect of the Stokes-Coriolis and Langmuir mixing parameterization is obtained on the Southern Ocean in winter, I suggest to mention the results of*

*these works in comparing the results of the VR12-AL-SC-EN experiment with respect to the CTRL one. Also Couvelard et al (2020) showed that there is a significant difference between annual averages of the module of the surface Stokes drift and of the part that is aligned with the wind (their Fig. 2). Please discuss.*

Response: We agree that the intensity of Langmuir cell strongly depends on the effect of misalignment. However, Li et al. (2016) found that the difference in a global coupled model between parameterizations with alignment and with misalignment was not significant, owing to the relatively coarse resolution. Our study has a relatively coarse resolution too, compared with Couvelard et al (2020). And the angles between winds and waves are less than 30° in most areas (Fig. S1i&j in the supplementary). So we didn't consider the effect of misalignment in the study. The results of previous works are introduced in Section 1, and the reasons for not considering the misalignment are added in text accordingly.

*Also, the description of the exchanges of the different parameters between model compartments is unclear to me. I understand that all additional terms are computed in WW3, and that the Stokes drift and Langmuir mixing terms are transferred to MOM4, that the Stokes drift is transferred to GFS for computed the surface relative wind, and that the Charnock parameter is also transferred to GFS for computing the surface roughness (Fig.1). What is unclear is what is exchanged between GFS and MOM4? Especially, are the (regular) surface currents transferred from MOM4 to GFS and used for estimating a relative wind velocity in computing the turbulent fluxes by GFS? If so, is it consistent with the transfer and use of the Stokes drift from WW3? Please provide the corresponding information, with an update of Fig.1. What is the meaning of the blue arrow from the coupler to GFS in Fig.1?*

Response: To clarify the exchanges between model components, we updated the original Fig. 1 as shown in Fig. R1. Between GFS and MOM4, originally GFS receives SST from MOM4, and sends fluxes of heat, momentum, and freshwater to MOM4 via mlc_coupler (black arrows). In the study, the (regular) surface currents from MOM4 are transferred to GFS via C_Coupler2 to estimate the relative wind velocity for the turbulent fluxes (Eqn. 11-13) in GFS. Similarly, the Stokes drift from WW3 is also transferred to GFS. The blue arrows in Fig. R1 indicate the surface currents transferred from MOM4 to GFS and WW3. The text in Section 2.1 and Fig. 1 are revised accordingly.

*About the effect of the surface current on the atmosphere: I guess from eq. 7 to 9 and section 2.4 that only the effect of the currents (and especially of the Stokes drift) on the turbulent fluxes is taken into account, and not the effect of the current on the surface wind through the tridiagonal matrix (see the work of Lemarié 2015). If so, the fact that the coupling is not complete should be clearly stated in section 2.4.*

Response: To complete the coupling, we add the surface current and Stokes drift to the tridiagonal matrix (Lemarié 2015) in CFS for the FLUX experiment. The experiments are re-run and the associated figures and text are revised.

[Figure]

**Figure R1.** A schematic diagram of the atmosphere-ocean-wave coupled modeling system. The arrows indicate the coupled variables that are passed between the model components. In the diagram, $C_{ch}$, $La_t$, $u_s(0)$, $V_s$, $U_{10}$, and $U_{surf}$ are Charnock parameter (red arrows), turbulent Langmuir number (red arrows), surface Stokes drift velocity (red arrows), Stokes drift transport (red arrows), 10-m wind (green arrows) and surface current (blue arrows), respectively.

*2- The statistical analysis of the different sensitivity tests is much clearer and easier to understand than in the previous version of the paper. I still feel rather uncomfortable with the different diagnostics used by the authors. For instance, the correlation between the bias reduction and the absolute bias shown in Fig. 3,4, 8 to 11 is almost never commented, and I am not sure about its meaning: from the text, I guess that its corresponds to the correlation between the relative change between CTRL and ALL (the so-called PRD) and the absolute bias, but only when the time evolution of the bias corresponds to an increase. Is it so? What is the additional information with respect to the PRD as shown elsewhere? Please elaborate.*

Response: We apologize for the confusion. In the revised text, we deleted these confusing figures and replace them with the time series of RMSEs in five experiments (e.g. Fig. R2b-Fig.R4b). The text is revised accordingly.

*For most of the parameters compared in this study, the maps represent the relative improvement (PRD). For the 10-m wind speed and MLD however, differences with the CTRL are given and I find these maps easier to read. Please justify why you use different diagnostics or homogenize. The relative improvement (PRD) depends strongly on the initial value of the bias. Why not showing maps of the biases for the different simulations? It would help to appreciate where the biases have been corrected or not. Please give the values of the final biases (and RMSE) for every parameter/experiment, in addition to the PRD.*

Response: To clearly show the changes of biases in the sensitivity experiments, we have replaced all maps of PRD with the differences relative to the CTRL. The global averaged biases and RMSEs are shown in the upper right of each map (e.g.Fig.R2-R4 for boreal winter). The figures for boreal summer are also updated. The Section 4 is revised accordingly.

[Figure]

**Figure R2.** The 53-day average SST (°C) bias in CTRL (a; CTRL minus OISST), the time series of global-averaged RMSE (b), and the differences between VR12-AL-SC-EN (c)/Z0-M04 (d)/ FLUX (e)/ ALL (f) and CTRL in Jan-Feb, 2017 (VR12-AL-SC-EN/Z0-M04/FLUX/ALL minus CTRL). The first 3-day simulation is discarded. The dotted areas are statistically significant at 95% confidence level.

[Figure]

**Figure R3.** The same as Figure R2 but for WSP10 (m/s).

[Figure]

**Figure R4.** The same as Figure R2 but for SWH (m).

*The comparison of the 10-m wind speed and SWH with the NDBC follows some of my previous recommendation, and I thank the authors for that. I think, however, that the way this comparison is presented could be greatly improved. I suggested that maybe, the wind speed can influence the bias and the difference between CTRL and ALL, and this comment is still valid. There is some effect of the value of the bias with CTRL, even though the current presentation of the results makes it difficult to apprehend. Rather than a table giving the relative difference for different quantiles of biases, I would suggest using a graph comparing directly the 10-m wind speed of the simulation outputs (y-axis) with the 10-m wind speed of the NDBC buoys (x-axis) in wintertime (same in summertime, and for the SWH), every dot on the graph representing a buoy (4 graphs in total). The results of the different simulations can be plotted in the same graph, with different colors. This would enable a direct comparison, including the effect of the wind speed (x-axis) and of the bias (distance to the y=x line). The changes between the different simulations can be given by the mean biases and standard deviations with respect to observations, rather than the relative mean changes.*

Response: As suggested, a graph comparing the 10-m wind speed and SWH from simulations with the NDBC buoys are shown in Fig. R5 (locations shown in Fig. S3, buoy identifiers with total numbers, longitudes and latitudes listed in Table S3). The x-axis is WSP10/SWH of buoys, and the y-axis is the simulated WSP10/SWH in Jan-Feb, 2017 (Fig. R5a&b) and Aug-Sep, 2018 (Fig. R5c&d) for all experiments. The mean biases with standard deviations and RMSEs for every experiment are shown in Table R1. The differences between 4 sensitivity experiments and CTRL are all statistically significant at 95% confidence level. Compared with NDBC data, the WSP10 and SWH in CTRL are generally overestimated in both winter and summer with positive mean biases (Fig. R5 & Table R1). Except the FLUX experiment in boreal winter, all other experiments show reduction of mean biases. The wave-related processes are most effective in areas with positive biases, consistent with previous comparisons with ERA5. As shown in Fig. R5, with the increase of

WSP10s and SWHs, the reduction of overestimation in ALL compared with CTRL is more prominent. The text in Section 4.3 is revised.

[Figure]

**Figure R5.** Scatter plots of simulated WSP10/SWH (y-axis) vs buoy WSP10/SWH (x-axis): (a) the WSP10 in Jan-Feb, 2017, (b) the SWH in Jan-Feb, 2017, (c) the WSP10 in Aug-Sep, 2018, and (d) the SWH in Aug-Sep, 2018. The dotted line is y=x. The corresponding mean biases with standard deviations and RMSEs for every experiment are shown in Table R1.

**Table R1.** The 53-day mean bias with standard deviation (STD) and RMSE for WSP10 and SWH compared with NDBC buoy observation: the bias is calculated as simulation minus NDBC.

| Boreal Winter WSP10 | Bias with STD | RMSE |
|:---:|:---:|:---:|
| CTRL | 0.16±1.23 | 1.24 |
| VR12-AL-SC-EN | 0.01±1.12 | 1.12 |
| Z0-M04 | -0.01±1.07 | 1.07 |
| FLUX | 0.39±1.20 | 1.26 |
| ALL | 0.07±1.04 | 1.04 |
| **Boreal Winter SWH** | **Bias with STD** | **RMSE** |
| CTRL | 0.21±0.38 | 0.44 |
| VR12-AL-SC-EN | 0.14±0.35 | 0.37 |
| Z0-M04 | 0.10±0.30 | 0.32 |
| FLUX | 0.24±0.34 | 0.42 |
| ALL | 0.12±0.34 | 0.36 |

| Boreal Summer WSP10 | Bias with STD | RMSE |
| --- | --- | --- |
| CTRL | 0.15±1.23 | 1.24 |
| VR12-AL-SC-EN | -0.03±1.22 | 1.22 |
| Z0-M04 | -0.04±1.21 | 1.21 |
| FLUX | -0.22±1.18 | 1.20 |
| ALL | -0.17±1.14 | 1.15 |
| Boreal Summer SWH | Bias with STD | RMSE |
| CTRL | 0.28±0.25 | 0.38 |
| VR12-AL-SC-EN | 0.19±0.24 | 0.30 |
| Z0-M04 | 0.22±0.26 | 0.34 |
| FLUX | 0.14±0.25 | 0.29 |
| ALL | 0.12±0.21 | 0.24 |

*3- Interpreting the results could, again, be made in a more accurate and concise way. For instance, the discussion in section 4.3 is rather long and not very easy to follow. Probably the effects in boreal summer with respect to boreal winter could be presented more briefly.*

Response: We modified the Section 4, especially the Section 4.3, to make the discussion more concise.

*Overall, I am not sure commenting in details improvements of a few percent is meaningful, but adding information about the correlation between the changes of different parameters can help to interpret the results. For instance, what is the correlation between the 2-m temperature and the SST changes? It confirm that the SST change is actually at the origin of the 2-m temperature change. Also, the correlation between the bias changes in 10-m wind speed and SWH is probably high. Please give values and discuss.*

Response: As suggested, the correlation coefficients between the SST and the T02 changes (ALL minus CTRL) in boreal winter and summer are calculated. The values are 0.61 and 0.53 respectively, significant at 99% confidence level, indicating the SST change is actually the origin of the 2-m temperature change. The correlation coefficients between WSP10 difference and SWH difference in ALL compared with CTRL are 0.77 and 0.73 in boreal winter and summer, significant at 99% confidence level, indicating the SWH change is originated from wind speed. The text is revised accordingly in Section 4.1 and 4.3.

*At some parts, the interpretation is not complete. For instance, the part about the latent heat flux does not lead to clear, concise results.*

Response: Indeed. Since the part about the latent heat flux does not produce significant results, we removed this part.

*From Fig. S3, I understand that the time evolution of the absolute value of the biases of the different parameters considered is overall positive, both in winter and in summer. At most places, these trends are significant, and even large, like more than 0.02°C/day (corresponding to more than 1°C difference for the 53-day simulated period) or 0.02 m/day for SWH (more than 1 m difference). What is the implication for the mean biases, and their changes from CTRL to ALL? Does it mean that the simulations are drifting in time from their initial state, because there is no assimilation of data? Or,*

*conversely, that their stationarity is not reached yet because the simulated period are too close from the initial state, despite the hot start? Please comment on that.*

Response: Yes, the simulations are drifting in time from their initial state because there is no data assimilation. From Fig. R2b-R4b, the RMSEs of SST, WSP10 and SWH increase in the first several weeks and then level off. So the simulated periods are not too close from the initial state.

*I specifically asked about the possible effects of including the parameterization of processes related with waves on the turbulent heat fluxes. The authors added a section about that effect, and I thank them for investigating it. However, it appears not significant, at least not for the time scales considered in this study.*

Response: The turbulent heat fluxes change is primarily resulted from the change of 10-m wind speed, and it is not significant due to the relative short simulation period. So the related discussion is removed in the text.

*Detailed comments*

*l. 26-29: are you sure that the SST change is at the origin of the 10-m wind speed change? I understand from 4.3 that the change of z0 also plays a role. Please check.*

Response: We apologize for the confusion. The corresponding lines are changed as "For WSP10s and SWHs, the wave-related processes generally lead to reduction of biases in regions where WSP10s and SWHs are overestimated. On one hand, the decreased SSTs stabilize marine atmospheric boundary layer, weaken WSP10s and then SWHs. On the other hand, the increased roughness length due to waves leads to reduction in the originally overestimated WSP10s and SWHs. In addition, the effects of Stokes drift and current on air-sea fluxes also rectify WSP10s and SWHs" in abstract.

*l. 65-66: the studies cited here are at climate scale, not for numerical prediction. The only model including wave effects and used for numerical prediction is ECMWF (IFS-WAM).*

Response: We agree. The related text is revised.

*l. 112-114: this set of experiment follows one of my previous question (is there any impact of the coupling frequency?). Table S1 brings some statistics but they are not commented in the text. Please justify why 1800s was chosen as the coupling frequency.*

Response: The coupling frequency indeed influences the simulation results slightly. To quantify, the RMSEs of SST, SWH and WSP10 with different coupling steps for the ALL experiment are shown in Table R2. From Table R2, the 10_STEP_WW3 experiment has the closest RMSEs to the 1_STEP_ALL for SST, SWH and WSP10, and has the relatively small runtime. Therefore, the time steps of 10_STEP_WW3 are selected to compromise computing time consumption and the model RMSEs. Text is revised to clarify in Section 2.1.

**Table R2.** The 28-day global RMSEs and daily runtime for SST, SWH and WSP10 in the ALL experiment with different coupling steps in Jan, 2017. In 1_STEP_ALL experiment, three model components are coupled every time step, and in 5_STEP_ALL (10_STEP_ALL) they are coupled every 5 (10) steps. Particularly, in 10_STEP_WW3, only the WW3 is coupled every 10 time steps,

whereas the GFS and the MOM4 remain the one time step coupling frequency as the original settings in CFS. Note that the original one time step in CFS is 180 s, and 10 time steps are 1800 s.

| Experiments | SST (°C) | SWH (m) | WSP10 (m/s) | Daily runtime (s) |
|---|---|---|---|---|
| 1_STEP_ALL | 0.88 | 1.09 | 3.94 | 25677 |
| 5_STEP_ALL | 0.88 | 1.09 | 3.88 | 19546 |
| 10_STEP_ALL | 0.88 | 1.23 | 4.25 | 19012 |
| 10_STEP_WW3 | 0.88 | 1.10 | 3.95 | 19171 |

*l. 122-124: "the daily initial fields at 00:00 UTC.." I guess it is rather the initial field of the first day of each experiment? Or is the model re initialized every 24h from the operational analysis? Please specify.*

Response: Yes, the initial field here refers to the first day of each experiment, not re-initialized every 24h from the operational analysis. The text is revised to clarify.

*l. 285 and following: Couvelard et al (2020) also obtained improvement of SST/MLD biases in the Southern Ocean. Please discuss your results against theirs.*

Response: In addition to the modification of Charnock parameter, the Stokes drift-related forces and the Langmuir cell, Couvelard et al. (2020) also considered the wave-supported stress, the modified turbulence kinetic energy, as well as the misalignment of wind and waves for Langmuir cells. In their work, the SST overestimations and MLD underestimations are reduced mainly due to the modified turbulence kinetic energy scheme. In our system, we include the effects of Stokes–Coriolis force and entrainment (VR12-AL-SC-EN) to enhance mixing, and subsequently improve SST/MLD. Text is added in Section 4.1 to clarify the differences between our results and theirs.

*l. 315-316: "generally consistent", please quantify.*

Response: To quantify, the spatial correlation coefficients between the SST and the T02 changes in ALL relative to CTRL are calculated, which are 0.61 and 0.53 in boreal winter and summer respectively, significant at 99% confidence level.

*l. 332 and following: please give the correlation coefficient between ALL MLD and the observations, so they can be compared with the 0.55 value given for CTRL.*

Response: The correlation coefficient of MLDs south of 45ºS (north of 45ºN) with Argo observations in CTRL is 0.55 (0.68), while in ALL the correlation coefficient of MLDs south of 45ºS (north of 45ºN) enhances to 0.63 (0.78) in boreal winter (summer). Text is revised accordingly.

*l. 348: I do not know how to interpret the "negative trends of bias". What is the meaning of that?*

Response: We apologize for the confusion. We updated Figure 8b&9b to time series of RMSEs. The "negative trends of bias" is deleted.

*l. 349-351: I would rather say the opposite: the biases in SWH are directly related to the biases of the 10-m wind speed.*

Response: Text is revised as suggested to "The correlation coefficients between changes of WSP10s and changes of SWHs in ALL are 0.77 and 0.73 in boreal winter and summer respectively (Fig. 8f&10f and Fig. 9f&11f), significant at 99% confidence level, indicating that the SWHs changes

are closely related to changes of wind speeds".

*l. 390 and following: see general comments. A graph showing the biases (model vs obs) in the different simulations would be easier to understand. Also, listing the buoy numbers in the SI is probably not useful, especially without additional information (position, number of observations). Please indicate, for every comparison, the number of buoys used.*

Response: As suggested, we replace Table 2 with Fig. R5 and Table R1. The number of buoys used for every figure and the corresponding buoy identifiers with longitudes and latitudes are listed in the revised supplementary (Table S3).

*Section 4.4: investigating the heat exchanges is nice at climate scale, but probably not relevant at the time scale of 2 months. I asked to authors to check for that, but it seems that the latent heat flux is directly influenced by the 10-m wind. Plus, the discussion in this part does not lead to clear results (to me). What would be your conclusion, beyond "the latent heat flux depends on the wind speed only"?*

Response: Yes, we only concluded that the latent heat flux change depends on the wind speed. Since the wave-related effect on the turbulent heat fluxes is non-significant for the 2-month simulation, we remove the Section 4.4.

References

Bao, Y., Song, Z., & Qiao, F. (2020). FIOâ□□ESM version 2.0: Model description and evaluation. Journal of Geophysical Research: Oceans, 125, e2019JC016036. https://doi.org/10.1029/2019JC016036

Couvelard, X., Lemarié, F., Samson, G., Redelsperger, J. L., Ardhuin, F., Benshila, R., & Madec, G. (2020). Development of a two-way-coupled ocean–wave model: assessment on a global NEMO (v3. 6)–WW3 (v6. 02) coupled configuration.GMD, 13(7), 3067-3090.

Lemarié, F., 2015: Numerical modification of atmospheric models to include the feedback of oceanic currents on air–sea fluxes in ocean–atmosphere coupled models. INRIA Grenoble-Rhône-Alpes Tech. Rep. RT-464, 10 pp. [Available online at https://hal.inria.fr/hal-01184711/document.]

Li, Q., Webb, A., Fox-Kemper, B., Craig, A., Danabasoglu, G., Large, W. G., & Vertenstein, M. (2016). Langmuir mixing effects on global climate: WAVEWATCH III in CESM. Ocean Modelling, 103, 145-160.

Li, Q., Fox-Kemper, B., Breivik, Ø., & Webb, A. (2017). Statistical models of global Langmuir mixing.Ocean Modelling, 113, 95-114.

Polonichko, V.: Generation of Langmuir circulation for nonaligned wind stress and the Stokes drift, J. Geophys. Res., 102, 15773– 15780, https://doi.org/10.1029/97JC00460, 1997.

Shimura, T., N. Mori, T. Takemi, and R. Mizuta (2017), Long-term impacts of ocean wavedependent roughness on global climate systems, J. Geophys. Res. Oceans, 122, 1995–2011, doi:10.1002/2016JC012621.

Torres, O., Braconnot, P., Marti, O., & Gential, L. (2019). Impact of air-sea drag coefficient for latent heat flux on large scale climate in coupled and atmosphere stand-alone simulations. Climate Dynamics, (3), 2125-2144.

Van Roekel, L. P., Fox-Kemper, B., Sullivan, P. P., Hamlington, P. E., and Haney, S. R.: The form and orientation of Langmuir cells for misaligned winds and waves, J. Geophys. Res., 117, C05001, https://doi.org/10.1029/2011JC007516, 2012.

We sincerely appreciate the reviewer for her/his constructive comments on the manuscript. Our responses are listed as follows in blue. Text are revised accordingly.

**Review from Referee #2**

*Here are my comments.*
*Lines 150, 152 and 157 Van -> Van Roekel*
Response: As suggested, the text is revised.

*Line 169, section 2.3.2: the Stokes drift should also be used in the advection of any tracer, including temperature and also in the calculation of the vertical velocity in difference/convergence term NEMO4.*
Response: We agree. According to the work of Couvelard et al. (2020), we calculate the Stokes drift profile and add it to the corresponding advection terms and convergence terms in MOM4. The text in Section 2.3 is revised, and the results are updated.

*Line 179, section 2.4: there is an inconsistency in considering the impact of the surface current and of the surface Stokes drift on the momentum flux in the atmosphere model, but not in the wave model. With ST4 (and ST3), the surface momentum balance is re-evaluated in order to determine the friction velocity that is then used as part of the source terms calculation, and hence the evolution of the wave field. To be consistent, WW3 should be forced not with the absolute 10m wind, with the relative 10m wind with respect to the surface current.*
Response: As suggested, the relative 10-m wind with respect to the surface current is applied in the wave model as well. Figure R1 and text are updated accordingly.

[Figure]

**Figure R1.** A schematic diagram of the atmosphere-ocean-wave coupled modeling system. The arrows indicate the coupled variables that are passed between the model components. In the diagram, $C_{ch}$, $La_t$, $u_s(0)$, $V_s$, $U_{10}$, and $U_{surf}$ are Charnock parameter (red arrows), turbulent Langmuir number (red arrows), surface Stokes drift velocity (red arrows), Stokes drift transport (red arrows), 10-m wind (green arrows) and surface current (blue arrows), respectively.

*Line 216: as far as I can understand from the text, the ST4-Fan scheme is used for z0 in the*

*atmospheric model only, and not in WW3. This is not consistent and should be made clearer that z0 and hence u\* inside WW3 will still be based on a Charnock determined from a modified version of Janssen wave induced stress (Ardhuin et al. 2010).*

Response: To clarify, we revise Section 2.5.2 to address that the new z0 scheme (previously ST4-Fan and now revised to ST4-M04) is used in CFS, while the z0 in WW3 is calculated by the ST4 source term (Ardhuin et al., 2010) with the method of wave-induced kinematic stress (Janssen 1989, 1991).

*Also, WW3 can be run with a cap on z0, and hence on the Charnock it could return to the atmosphere. See z0max in table 2.6 in the WW3 manual. It is indeed set to a large value for TEST473f, however, TEST500 has z0max=0.002 for instance. I believe, it is worth mentioning as ST4-Fan is not the only way to limit Charnock for high winds. One should also mention some recent developments on modifying ST4 for high winds (Bidlot et al. 2020 and Li et al. 2021), without a very awkward parameterization of Fan et al. (sorry I notice that it is mentioned later (line 488), but it might too late).*

Response: As suggested, the introduction of "Modifications to these Charnock parameterizations were suggested in recent studies for the leveling off roughness under high winds (e.g. Fan et al., 2012; Bidlot et al., 2020; ECMWF, 2020; Li et al., 2021)" is moved to Section 1.

*I have serious problem with ST4-Fan. It will indeed limit Charnock for large winds, but from figure S2, it does not seem to make sense that Charnock is largest in the Tropics. The Charnock parameter was introduced to represent the impact of waves on the momentum transfer at the sea surface. It was recognized that young wind-sea should extract more momentum than older more mature old sea. So why is Charnock largest in the Tropics where the sea state should be dominated by old wind-sea and swell?*

Response: We agree that the calculation of Charnock parameter in ST4-Fan is problematic for the sea state dominated by old wind-sea and swell. To solve this problem, we revise ST4-Fan based on Moon et al. (2004). According to Fan et al. (2012), the equations of Charnock parameter are derived based on the observations in Moon et al. (2004). In Moon et al. (2004), the Charnock parameter decreases with the increase of wave age at low-middle winds (<30 m/s), but levels off or increases at high winds (>30 m/s; Fig. 3 of Moon et al. 2004). Moon et al. (2004) proposed Eqn. R1 to estimate the Charnock parameter, and gave different constant values of $a$ and $b$ changing with 10-m wind speed every 5 m/s in the range of 10 m/s to 50 m/s (Table 1 of Moon et al. 2004). Based on this, Fan et al. (2012) proposed Eqn. R2 to calculate $a$ and $b$. Because $b$ is always positive (Eqn. R2), the Charnock from Eqn. R1 increases with wave age even at low wind speed, which generates large Charnock in the tropics.

$$C_{ch} = a\left(\frac{c_{pi}}{u_*}\right)^b, \qquad\qquad (R1)$$

$$a = \frac{0.023}{1.0568^{U_{10}}}, b = 0.012 U_{10}, \qquad\qquad (R2)$$

Therefore, we have re-derived the relationship (Eqn. R3) between $a/b$ and 10-m wind speed $U_{10}$ by fitting the values in Table 1 of Moon et al. (2004),

$$a = \frac{1}{0.1477 U_{10}^2 - 0.7395 U_{10} - 10.9995}, \qquad\qquad (R3)$$

$$b = 1.5661E^{-5}U_{10}{}^3 - 0.002U_{10}{}^2 + 0.1017U_{10} - 1.6182.$$

In Eqn. R3, $b$ is negative (positive) from relatively small (large) wind speed. Because the observations in Moon et al. (2004) were obtained under tropical cyclones, Eqn. R3 is used for $U_{10}$>15 m/s, whereas the original Charnock relationship of WW3 ST4 scheme (Janssen 1989, 1991) is used for $U_{10} \leq 15$ m/s. The revised parameterization is called ST4-M04. The comparison of $z_0$ among ST4-M04, ST4-Fan, ST4 and GFS is shown in Fig. R2, indicating that the new ST4-M04 is close to ST4-Fan at high wind speed and can improve $z_0$ at low wind speed. Consequently, the Charnock parameter in ST4-M04 is low in the Tropics as expected (Fig. R3).

[Figure]

**Figure R2.** Relationships between momentum roughness length $z_0$ (m) in the coupled system and 10-m wind speed (m/s); error bars indicate twice the standard deviations for each point.

[Figure]

**Figure R3.** The Charnock parameter $C_{ch}$ obtained by ST4-M04 in boreal winter (a) and summer (b).

*Line 231: it is more than just the fetch, but also the development stage of the sea state (young sea, old sea…)*

Response: As suggested, the text is revised to "Since the $z_0$ is determined only by wind-sea conditions in ST4 and ST4-M04 scheme, the STD at a given wind speed is mainly owing to variations in wind fetch and development stage of sea state. The reduced STDs in ST4-M04 scheme, compared to ST4, imply less sensitivity of $z_0$ to fetch and sea state".

*Line 233, before section 2.6: Could you say what is done in WW3 when sea ice is present. It seems that there is a large impact near or within the sea ice, so it is important to know how the different fields were produced by WW3. Quite often, WW3 is set-up so that the wave spectra are reset to 0 every time step for all areas with a sea ice cover above a certain threshold. If it is the case, this will mean that the estimate for Stokes drift and wave age would be those for very young windsea. It can then be debated whether these estimates are correct, as it is known that in the presence of sea ice, the high frequency waves are heavily damped.*

Response: In WW3, the ice blocking IC0 source term with the critical ice concentration of 50% is applied. So for ice concentration in the range of 0-50%, the wave spectra are not zero, while for ice concentration larger than 50% the wave spectra are zero. Indeed, the estimate for Stokes drift and wave age would be those for young wind-sea, inconsistent with the phenomenon that high-frequency waves are severely damped in the presence of sea ice. Since the interactions between

waves and sea ice are complicated and beyond the scope of the study, we turn off the coupling between WW3 and CFSv2.0 in areas with sea ice. Text is revised to clarify in Section 2.1.

*Line 256 and table S4: NDBC website reports its own buoy data, as well as buoy data from other NOAA agencies, such as the data from the TAO array, and a few other buoy data providers along the US. But it also reports surface observations from many coastal stations (i.e. on land, piers or towers) that are definitely not buoy. They also report observations from the oil and gas industry in the Gulf of Mexico. In table S4, anything that does not have a 5-digit identifier is probably not a buoy, and the oil and gas data from the Gulf of Mexico have an identifier like 423xx, 428xx. All those non buoy data should not be use in this analysis.*

Response: As suggested, all non-buoy data are removed. The text is revised accordingly.

*Line 261: section 4.1: I am surprised by the lack of impact of the FLUX experiment on the SST, in particular in the equatorial Pacific. My experience is that including the winds relative the surface current in the momentum flux (surface stress) used to force the ocean model has quite an impact on the SST around the equatorial Pacific. To be sure, could you confirm, as indicated that MOM4 is indeed forced with the surface stress as shown in equation (7).*

Response: We checked the code as suggested, and the MOM4 is indeed forced with the surface stress as original Eqn. 7 (Eqn. 11 in revised text). Previously, we showed the SST changes by a percentage relative differences (PRD $= \frac{|\widehat{y_s}-y|-|\widehat{y_c}-y|}{|y|} \times 100\%$, where $y$ is OISST, $\widehat{y_c}$ is simulated SST in CTRL and $\widehat{y_s}$ is simulated SST in other experiments) in Fig. 3&4. Since the absolute value of OISST on the denominator is relatively large, the PRD looks low in lower latitudes. In the revised text, all the maps of PRD are replaced by SST differences between sensitive experiments and CTRL (Fig. R4&R5). The SST differences between FLUX and CTRL around the equatorial Pacific are manifest (Fig. R4e&R5e).

[Figure]

**Figure R4.** The 53-day average SST (℃) bias in CTRL (a; CTRL minus OISST), the time series of

global-averaged RMSE (b), and the differences between VR12-AL-SC-EN (c)/Z0-M04 (d)/ FLUX (e)/ ALL (f) and CTRL in Jan-Feb, 2017 (VR12-AL-SC-EN/Z0-M04/FLUX/ALL minus CTRL). The first 3-day simulation is discarded. The dotted areas are statistically significant at 95% confidence level.

[Figure]

**Figure R5.** The same as Figure R4 but for Aug-Sep, 2018.

*Line 474: it not true, the SST bias in the Tropics was not reduced due to drag from swell. Breivik et al. (2015) considered the impact of sea state dependent Charnock, using the formulation from Janssen on the air-side surface stress, which was modulated further considering the momentum flux balance between wind input and wave breaking to determine the ocean-side stress that is then used to force the ocean. Moreover, the impact of wave breaking was also considered as input to the upper ocean mixing scheme (TKE). All these effects had an impact on the SST. What was not discussed in Breivik et al. (2015) is the impact of surface currents on the SST response in the Tropical Pacific as that effect has already been introduced in the ECMWF system years before. As I mentioned earlier, I am surprised by the lack of sensitivity on the CFS system to surface currents in the Tropics.*

Response: We apologize for not showing the influence of surface currents in the tropics clearly. As shown in the updated figures (Fig. R4e&R5e), surface current and Stokes drift can influence SST in tropics. The values of changes can be up to more than $\pm 0.5℃$ in the tropical Pacific in boreal summer. Text is revised to clarify the effects of surface current and Stokes drift on SST. The descriptions of Breivik et al. (2015) are revised as suggested.

References
Ardhuin, F., Rogers, E., Babanin, A. V., Filipot, J., Magne, R., Roland, A., Der Westhuysen, A. V., Queffeulou, P., Lefevre, J. M., and Aouf, L.: Semiempirical Dissipation Source Functions for Ocean Waves. Part I: Definition, Calibration, and Validation, Journal of Physical Oceanography, 40, 1917-1941, http://dx.doi.org/10.1175/2010JPO4324.1, 2010.
Breivik, O., Mogensen, K., Bidlot, J., Balmaseda, M., and Janssen, P. A. E. M.: Surface wave effects

in the NEMO ocean model: Forced and coupled experiments, Journal of Geophysical Research, 120, 2973-2992, http://dx.doi.org/10.1002/2014JC010565, 2015.

Couvelard, X., Lemarié, F., Samson, G., Redelsperger, J.-L., Ardhuin, F., Benshila, R., and Madec, G.: Development of a two-way-coupled ocean-wave model: assessment on a global NEMO(v3.6)-WW3(v6.02) coupled configuration, Geosci. Model Dev., 13, 3067-3090, https://doi.org/10.5194/gmd-13-3067-2020, 2020.

Fan, Y., Lin, S., Held, I. M., Yu, Z., and Tolman, H. L.: Global Ocean Surface Wave Simulation Using a Coupled Atmosphere–Wave Model, Journal of Climate, 25, 6233-6252, http://dx.doi.org/10.1175/JCLI-D-11-00621.1, 2012.

Janssen, P. A. E. M.: The Quasi-linear theory of wind wave generation applied to wave forecasting. J. Phys. Oceanogr. 21, 1631-1642. https://doi.org/10.1175/1520-0485(1991)021<1631:QLTOWW>2.0.CO;2, 1991.

Janssen, P. A. E. M.: Wave-induced stress and the drag of air flow over sea waves. J. Phys. Oceanogr. 19(6), 745-754. https://doi.org/10.1175/1520-1310 0485(1989)019<0745:WISATD>2.0.CO;2, 1989.

Moon, I., Ginis, I., and Hara, T.: Effect of surface waves on Charnock coefficient under tropical cyclones, Geophysical Research Letters, 31, http://dx.doi.org/10.1029/2004GL020988, 2004.

Saha, S., Moorthi, S., Wu, X., Wang, J., Nadiga, S., Tripp, P., Behringer, D., Hou, Y., Chuang, H., and Iredell, M. D.: The NCEP Climate Forecast System Version 2, Journal of Climate, 27, 2185-2208, http://dx.doi.org/10.1175/JCLI-D-12-00823.1, 2014.

---

## Referee Report (RR1)

**Interactive comments on The Effects of Ocean Surface Waves on Global Intraseasonal Prediction: Case Studies with a Coupled CFSv2.0-WW3, Shi et al., GMDD, https://doi.org/10.5194/gmd-2021-322**

I thank the authors for taking into account in a very careful way the comments of both reviewers. Especially, they significantly improved the coupling between the ocean, wave and atmospheric models by including the effects of the surface currents into the wind (and not only turbulent fluxes) and by changing the parameterization of the drag coefficients. Also, the discussion comparing the present study with previous results is more complete and the diagnostics of the biases with respect to observations and their evolution with different configurations has been much improved. I am convinced that they added significance and visibility to their results by making the present version of the paper more complete and clearer. Still, the language could be improved and the paper deserves a very careful spell checking (the suggestions below are not comprehensive). I only have minor comments as follows.

Title: a noun is missing after CFSv2.0-WW3, I suggest "CFSv2.0-WW3 System" or "CFSv2.0-WW3 configuration".
l. 50: for Cch → for defining Cch
l. 139: please provide information here about the different coupling configuration tested and the details corresponding to 10_STEP_WW3.
l. 145: Charnock parameter related estimation → estimation of the Charnock parameter
l. 146: offered → available
l. 154: Compared → Comparing
l. 164: The → Where
l. 178: supplementary → supplementary material or information
l. 180: varies in proportion to → depends on
l. 182: is the → the
l. 187: is an average value → is the average value of the density
l. 217: To account for the effects of Stokes drift velocity, the Eqn. 14 was applied → To account for the effects of the surface currents and of the Stokes drift, Eqn. 14 was used
l. 219: is also conducted → has been implemented
l. 221: with an angle → different
l. 226: Eqs 11-13 do not show the link between the roughness length and the transfer coefficients, please rephrase.
l. 284: applied → used
l. 288: sentence is not clear, please rephrase.
l. 303: This is different with → this contrasts with
l. 361: I don't understand the second part of the sentence; what is the meaning of perturbation here? In the "results" part, the authors should add a comment about the general increase of the biases (wrt ERA5 or observations) in all experiments, and the fact that it is likely a drift from the initial conditions (because no data are assimilated).

---

## Author Response (AR2)

**Review from Referee #1**

*I thank the authors for taking into account in a very careful way the comments of both reviewers. Especially, they significantly improved the coupling between the ocean, wave and atmospheric models by including the effects of the surface currents into the wind (and not only turbulent fluxes) and by changing the parameterization of the drag coefficients. Also, the discussion comparing the present study with previous results is more complete and the diagnostics of the biases with respect to observations and their evolution with different configurations has been much improved. I am convinced that they added significance and visibility to their results by making the present version of the paper more complete and clearer. Still, the language could be improved and the paper deserves a very careful spell checking (the suggestions below are not comprehensive). I only have minor comments as follows.*

Response: We sincerely appreciate the reviewer for her/his constructive comments on the manuscript. Our responses are listed as follows in blue. Text is revised accordingly.

*Title: a noun is missing after CFSv2.0-WW3, I suggest "CFSv2.0-WW3 System" or "CFSv2.0-WW3 configuration".*

Response: As suggested, "System" is added.

*l. 50: for Cch → for defining Cch*

Response: As suggested, "defining" is added.

*l. 139: please provide information here about the different coupling configuration tested and the details corresponding to 10_STEP_WW3.*

Response: The corresponding information is added as below, "The three components are coupled every time step (180 s) in 1_STEP_ALL experiment, every 5 steps (900 s) in 5_STEP_ALL experiment and every 10 steps (1800 s) in 10_STEP_ALL experiment. In 10_STEP_WW3, only the WW3 is coupled every 10 time steps, whereas the GFS and the MOM4 remain the one time step (180 s) coupling frequency as the original settings in CFSv2.0".

*l. 145: Charnock parameter related estimation → estimation of the Charnock parameter*
*l. 146: offered → available*
*l. 154: Compared → Comparing*
*l. 164: The → Where*
*l. 178: supplementary → supplementary material or information*
*l. 180: varies in proportion to → depends on*
*l. 182: is the → the*
*l. 187: is an average value → is the average value of the density*
*l. 217: To account for the effects of Stokes drift velocity, the Eqn. 14 was applied → To account for the effects of the surface currents and of the Stokes drift, Eqn. 14 was used*
*l. 219: is also conducted → has been implemented*
*l. 221: with an angle → different*

Response: Thanks. We have modified the manuscript according to all these comments.

*l. 226: Eqs 11-13 do not show the link between the roughness length and the transfer coefficients, please rephrase.*

Response: Text is revised to "The fluxes are in part determined by surface roughness length, which can be converted to surface exchange coefficients based on the Monin-Obukhov similarity theory (Monin and Obukhov, 1954)".

*l. 284: applied → used*

Response: Done. Thanks.

*l. 288: sentence is not clear, please rephrase.*

Response: Text is revised to "The results in the first three days were excluded in the evaluation, since the wave influences were weak at the beginning".

*l. 303: This is different with → this contrasts with*

Response: Done. Thanks.

*l. 361: I don't understand the second part of the sentence; what is the meaning of perturbation here? In the "results" part, the authors should add a comment about the general increase of the biases (wrt ERA5 or observations) in all experiments, and the fact that it is likely a drift from the initial conditions (because no data are assimilated).*

Response: We apologize for the confusion. The "with perturbations" is deleted. As suggested, a comment is added in Section 4 as below, "Compared with observations or ERA5, the general increase of the biases in all experiments is likely a drift from the initial conditions since no data are assimilated".

References

Monin, A. S., and Obukhov, A. M.: Basic laws of turbulent mixing in the surface layer of the atmosphere, Contrib. Geophys. Inst. Acad. Sci. USSR, 151(163), e187, 1954.

---

## Author Response (AR3)

**Review from Editor**

*Dear Author,*

*Thank you for your revised manuscript that carefully takes into account the remarks from the 2nd round of reviews. Your manuscript can be considered for publication after you take into account the following technical corrections.*

Response: We sincerely appreciate the editor for her constructive comments on the manuscript. Our responses are listed as follows in blue. Text is revised accordingly.

*L.26: « by Stokes drift » → « by the Stokes drift »*
*L.30: « the effects of Stokes drift » → « the effects of the Stokes drift »*
*L.45: « shear of Stokes drift » → « the shear of the Stokes drift »*
*L.50: « assessed from » → « considering »*
*L.62: « reaching maximum » → « reaching a maximum »*
*L.73: « has been released with great flexibility » → « has been assembled »*
*L.76: « it takes sufficient periods » → « it takes significant time »*
*L.77: « impact of individual wave process » → « impact of individual wave processes »*
*L.78-90: use present tense everywhere*
*L.82: « The CFSv2.0 is a coupled system with the main application for ... » → « The CFSv2.0 is a coupled system mainly applied for ... »*

Response: Thanks. We have modified the manuscript according to all these comments.

*L.84: the sentence on work at NCEP should be moved up at the end of the previous paragraph (L.74)*

Response: As suggested, "The National Centers for Environmental Prediction (NCEP) is establishing its atmosphere-ocean-wave system, in which the Global Forecast System (GFS; the atmosphere module in the Climate Forecast System model version 2.0) is one-way coupled with the WAVEWATCH III (WW3)" is moved to Line 74-76.

*L.105: « The MOM4 » → « MOM4 »*
*L.106: « with the enhanced horizontal resolution to 0.25°» → « with horizontal resolution enhanced to 0.25°»*
*L.121: « then generates and evolves » « generates »*
*L.149: « which include » → « which includes » (as I think it refers to the « energy balance equation »)*
*L.230: « the estimates of them are » → « their estimate is»*
*L.292: « results was presented. Comparisons were made » → « results is presented. Comparisons are made*

Response: Done. Thanks.

*On Figure 1, $u_S(0)$ appears twice from WAVEWATCH III and C-Coupler2*
Response: We apologize for the confusion. Figure R1 (Figure 1 in text) is updated accordingly.

[Figure]

**Figure R1.** A schematic diagram of the atmosphere-ocean-wave coupled modeling system. The arrows indicate the coupled variables that are passed between the model components. In the diagram, $C_{ch}$, $La_t$, $u_s(0)$, $V_s$, $U_{10}$, and $U_{surf}$ are Charnock parameter (red arrows), turbulent Langmuir number (red arrows), surface Stokes drift velocity (red arrows), Stokes transport (red arrows), 10-m wind (green arrows) and surface current (blue arrows), respectively.

*L.422: « CFSv2.0 for global oceans from 78oS-78oN » → « CFSv2.0 over the domain 78oS-78oN »*

*L. 436: « the smaller ocean » → « the relatively smaller ocean »*

*L.439: « are overestimated » → « were overestimated »*

*L.466: « The case studies » → « Our case studies »*

Response: Thanks. We have modified the manuscript according to all these comments.

*L.469: «What do you mean by « extra adjusting », please be more precise; at least, change « adjusting » for « adjustment »*

Response: We apologize for the confusion. Text is revised to "In addition, every individual model component could be further improved via new parameter settings or updated parametrization schemes".

*L. 470-471: « All these require further efforts to investigate efficient methods to improve fully coupled systems. » → «All these aspects will need to be considered for improving our coupled system. »*

Response: Done. Thanks.

*With very best regards*